# Proteome-wide 3D structure prediction provides insights into the ancestral metabolism of ancient archaea and bacteria

Weishu Zhao [1,7], Bozitao Zhong [1,2,7], Lirong Zheng [2], Pan Tan [2], Yinzhao Wang [1], Hao Leng[1], Nicolas de Souza [3], Zhuo Liu[2,4,5], Liang Hong [2,4,5] ✉ & Xiang Xiao [1,6] ✉

Ancestral metabolism has remained controversial due to a lack of evidence beyond sequence-based reconstructions. Although prebiotic chemists have provided hints that metabolism might originate from non-enzymatic proto-metabolic pathways, gaps between ancestral reconstruction and prebiotic processes mean there is much that is still unknown. Here, we apply proteome-wide 3D structure predictions and comparisons to investigate ancestorial metabolism of ancient bacteria and archaea, to provide information beyond sequence as a bridge to the prebiotic processes. We compare representative bacterial and archaeal strains, which reveal surprisingly similar physiological and metabolic characteristics via microbiological and biophysical experiments. Pairwise comparison of protein structures identify the conserved metabolic modules in bacteria and archaea, despite interference from overly variable sequences. The conserved modules (for example, middle of glycolysis, partial TCA, proton/sulfur respiration, building block biosynthesis) constitute the basic functions that possibly existed in the archaeal-bacterial common ancestor, which are remarkably consistent with the experimentally confirmed protometabolic pathways. These structure-based findings provide a new perspective to reconstructing the ancestral metabolism and understanding its origin, which suggests high-throughput protein 3D structure prediction is a promising approach, deserving broader application in future ancestral exploration.

Tracing and resurrecting ancestors from modern cells appears to be an indispensable approach for investigating the origin and evolution of early life on Earth, as the history of life is too long to find or interpret biological fossils dating back to the origin[1–3]. During the past two decades, many studies have proposed various ideas to reconstruct ancestral cells in silico. With the development of culture-independent high-throughput sequencing technology and bioinformatics approaches, sequence-based comparative genomic and phylogenomic

[1]State Key Laboratory of Microbial Metabolism, International Center for Deep Life Investigation (IC-DLI), School of Life Sciences and Biotechnology, Shanghai Jiao Tong University, 200240 Shanghai, China. [2]Institute of Natural Sciences, Shanghai National Center for Applied Mathematics (SJTU Center) and MOE-LSC, Shanghai Jiao Tong University, 200240 Shanghai, China. [3]Australian Nuclear Science and Technology (ANSTO), Locked Bag 2001, Kirrawee DC, Sydney, NSW 2232, Australia. [4]Shanghai Artificial Intelligence Laboratory, 200232 Shanghai, China. [5]School of Physics and Astronomy, Zhangjiang Institute for Advanced Study, Shanghai Jiao Tong University, 200240 Shanghai, China. [6]Southern Marine Science and Engineering Guangdong Laboratory (Zhuhai), Zhuhai, Guangdong, China. [7]These authors contributed equally: Weishu Zhao, Bozitao Zhong. ✉e-mail: hongl3liang@sjtu.edu.cn; zjxiao2018@sjtu.edu.cn

analyses are the major avenues for studying the origin and ancestry of life. The main challenge of phylogenomic analysis is to identify where the phylogenetic tree should be rooted and even whether the root should exist at all. The latest phylogenetic approaches have rooted archaeal and bacterial trees independently and reconstructed the potential lifestyle of the last archaeal common ancestor (LACA) and the last bacterial common ancestor (LBCA), respectively[4,5], furnishing valuable and important information on early life. The lack of additional evidence beyond sequence-based conclusions makes ancestral reconstruction controversial, especially for ancestral physiological and metabolic characteristics[6].

In a widely accepted view, the ancestor was a thermophilic anaerobe that originated from gasses in a hydrothermal setting[6–9], contrasting views notwithstanding[10,11]. Sequence-based metabolic reconstruction indicates a potential autotrophic or mixotrophic lifestyle for early cells, providing incomplete biosynthesis pathways for the building blocks for life, including some essential amino acids and cofactors[4,5]. In addition, it is generally assumed that cellular ancestors arose from building blocks (i.e., nucleic acids, amino acids, peptides, lipid vesicles) under 'prebiotically plausible' conditions and were underpinned by non-enzymatic prebiotic metabolism[12]. A recent thermodynamic study on conserved and universal core reactions agreed that metabolic pathways possibly originated in hydrothermal systems, in which 95–97% of the synthesized reactions are exergonic under the environmental conditions of hydrothermal vents[13]. Both sequence- and reaction-based investigations have indicated that ancient metabolism may have been a hybrid of biotic and abiotic processes. However, it remains unclear which metabolic reactions in thermophilic ancestral cells arose from the non-enzymatic prebiotic metabolism and how they were kept by enzymes and conserved during evolution.

The structures of biomolecules, particularly the 3D structures of proteins, can provide crucial information connecting genomic sequences and bio functions. Direct comparison of the structures of all proteins between the representative archaea and bacteria belonging to ancient groups might provide more intuitive and in-depth information on the functional commonality and difference between archaea and bacteria compared to genomics alone. Experimental approaches to obtain the 3D structure of thousands of proteins are challenging. However, the recent development of advanced deep learning methods, such as AlphaFold2, can deliver the atomic structure of proteins based on its 1D sequence with a high degree of accuracy[14].

In this work, we perform a proteome-wide comparison of 3D protein structure on two representatives of ancient archaeal and bacterial species that we isolated in previous studies (Fig. 1), as additional information beyond sequences to explore the potential ancestor of early archaea and bacteria. One is a hyperthermophilic archaeon, *Thermococcus eurythermalis* A501 (denoted A501), which belongs to a representative ancient group of archaea[4,15,16], and the other is a newly discovered and isolated thermophilic bacterium, *Zhurongbacter thermophilus* 3DAC (denoted 3DAC), which belongs to a novel phylum-level lineage and provides new insights into an early-diverging bacterial cluster (superfamily Zhurongbacteria) with genomic and physiological features that have never been reported in any other bacteria[17] (Fig. 1B). Both were isolated from deep-sea hydrothermal vents, which are believed to be one of the cradles of life[18], from different locations. Apart from temperature adaptation, these two strains showed similar habitats, physiological and metabolic characteristics, and thermal resilience (Fig. 1D, Supplementary Note, Supplementary Fig. 1), which avoids the general challenge of the low comparability between archaea and bacteria restricted by greatly different ecological niches[4]. According to conserved or variable protein structures of a series of enzymes between A501 and 3DAC, metabolic pathways could be spontaneously distinguished into distinct modules, which are further extended and confirmed in 24 other representative archaea and bacteria with diverse positions across the phylogenetic tree. Interestingly, we find the conserved metabolic modules in central carbon metabolism predominated by highly conserved protein structures among archaea and bacteria are exactly consistent with the experimentally confirmed protometabolic pathways by prebiotic priocesses[19–29], which provides a new perspective to reconstruct the ancestral metabolism that may present in archaeal-bacterial common ancestor (ABCA) and understand the origin of metabolism.

## Results

### Physiology and metabolism of A501 and 3DAC

Strain A501 was obtained from the Guaymas Basin[15], while strain 3DAC was obtained from the East Pacific Ridge (EPR). The main difference between these two strains was the growth temperature (Fig. 1C). Strain 3DAC performs as a thermophile, in which the growth temperature ranges from 30 to 75 °C with an optimal temperature of 70 °C. In contrast, A501 acts as a hyperthermophile that grows from 50 to 100 °C, with an optimal temperature of 85 °C. In other words, A501 has a higher optimal temperature and a wider growth temperature range than 3DAC. In addition, A501 has a higher specific growth rate under optimal, suboptimal, or superoptimal temperatures (Fig. 1C). Except for the growth temperature, other physiological (e.g., pH, salinity, pressure) and metabolic features of A501 and 3DAC are quite similar (Supplementary Note). Both A501 and 3DAC are strictly anaerobic heterotrophs that rely on supplementary essential amino acids, utilize carbohydrates or peptides (amino acids) as carbon sources, and use sulfur as an energy source to stimulate growth (Supplementary Note, Supplementary Fig. 1). Growth experiments on various carbon sources revealed that eight amino acids are essential for the growth of A501, while 11 are essential for 3DAC, among which seven essential amino acids overlap between the two species (i.e., L-arginine, L-histidine, L-methionine, L-phenylalanine, L-tyrosine, L-tryptophan, and L-threonine). In the presence of amino acids, carbohydrates such as glucose can be used as carbon sources (see the "Methods" section, Supplementary Note, Supplementary Fig. 1). These wet-lab experimental results suggest that both carbon and energy metabolism are comparable between A501 and 3DAC.

### Comparison of the thermal resilience of biomacromolecules by neutron scattering

Considering the key difference in temperature adaptation between A501 and 3DAC, we first estimated the thermal resilience of biomacromolecules in their cells, which was quantitatively determined by the slope of the mean squared atomic displacement (MSD, $\langle x^2(\triangle t)\rangle$) as a function of temperature[30] derived from neutron scattering as $\langle k'\rangle = 0.00276/(\mathrm{d}\langle x^2(\triangle t)\rangle/\mathrm{d}T)$ (see the "Methods" section), where the steeper the slope is, lower the mechanical resilience against the thermal perturbation[30] (Fig. 1D). For a simple model of an elastic spring, the so-defined resilience is nothing but the elastic constant, which directly measures the stiffness of the spring. Many research works have used this method to measure the stiffness of protein molecules[31–37] or the average stiffness of the biomolecules inside the cells[30,38–44]. The mean resilience values of biomolecules in A501, 3DAC, and *E. coli* were found to be 0.78, 0.67, and 0.31 N/m, respectively (see the "Methods" section). The resilience of A501 is slightly higher than that of 3DAC, but both showed much higher values than *E. coli*. The higher resilience may help the biomacromolecules of A501 and 3DAC adapt to the significant temperature changes in deep-sea hydrothermal vents, and indicate a general similarity of biomacromolecule thermostability between A501 and 3DAC.

### Proteome-wide comparison of structures between A501 and 3DAC

We further predicted and compared protein 3D structures at the proteome scale between A501 and 3DAC, as proteins constituted the

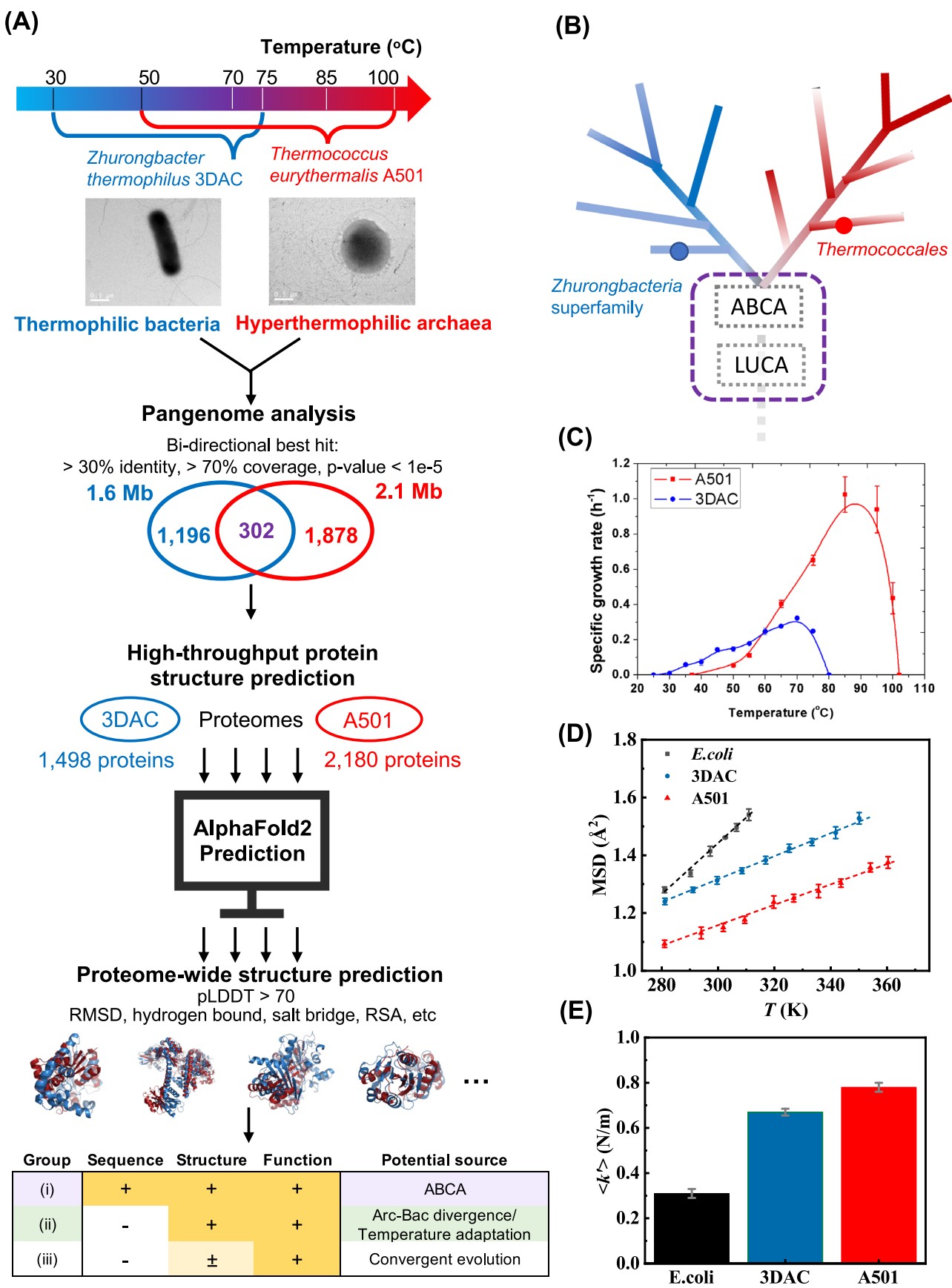

**Fig. 1 | Workflow and basic characteristics of A501 (red) and 3DAC (blue) in this study. A** Schematic diagram of the AlphaFold2-based proteome-wide structure prediction was used in this study. **B** Phylogenetic positions of the Zhurongbacteria superfamily and Thermococcales to which 3DAC and A501 belong. "+" represents conserved in ortholog or structure or function; "−" represents not conserved; "±" represents in some cases the active site is conserved but in most cases not. "ABCA" represents the archaeal-bacterial common ancestor. "Arc-Bac divergence/temperature adaptation" means independent evolution after the division of archaea and bacteria or according to temperature adaptation. "LUCA" represents the last universal common ancestor. **C** Growth features in the temperature range of A501 and 3DAC. Error bars represent the standard deviations (SDs) from independent

biological triplicates ($n = 3$) during the experiments under 0.1 MPa. Data are presented as average values ± SD. **D** Mean square displacement, $\langle X^2 (\Delta t)\rangle$, of biomacromolecules in strains A501 and 3DAC as a function of temperature in vivo compared with a mesophilic model strain *Escherichia coli* (*E. coli*). **E** Mean macromolecular resilience $\langle k \rangle$ for *E. coli*, 3DAC and A501. The $\langle k \rangle$-values of *E. coli*, 3DAC and A501 is $0.31 \pm 0.020$, $0.67 \pm 0.015$, $0.78 \pm 0.022$ N/m, respectively. The $\langle k \rangle$ errors were calculated from the slope of the weighted straight-line fits to the mean squared atomic displacement (MSD) data by using the Levenberg–Marquardt algorithm[30]. Biologically independent samples ($n = 3$) were mixed to examine over one independent experiment for neutron scattering. Source data are provided as a Source Data file.

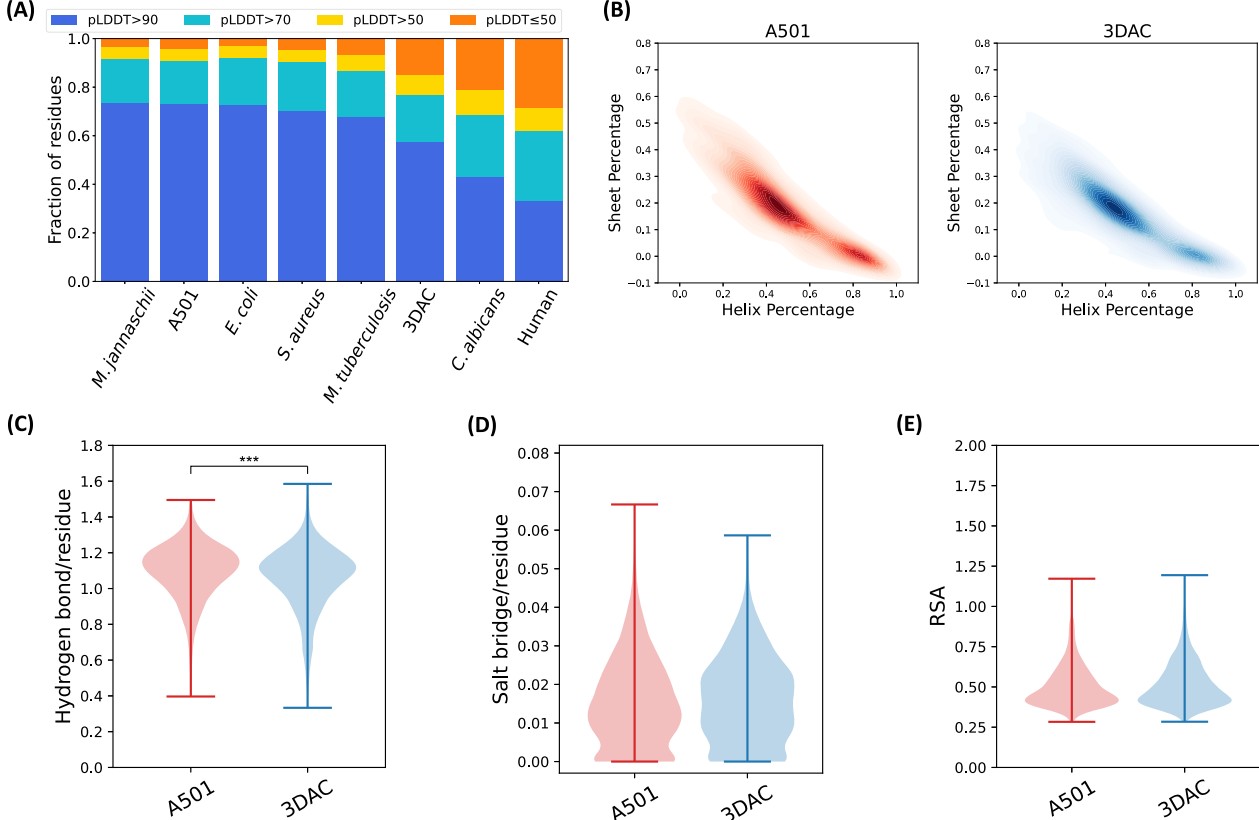

**Fig. 2 | Construction of proteome-wide protein structures and statistical comparison between A501 (red) and 3DAC (blue). A** Composition of average pLDDT per residue in proteome-wide protein structures in A501 and 3DAC compared with the model organisms. **B** Comparison of helix-sheet percentage between A501 and 3DAC. Comparison of hydrogen bonds (**C**), salt bridges (**D**), and relative surface areas (**E**) per amino acid residue in all proteins of A501 ($n = 2180$) and 3DAC

($n = 1498$). Hydrogen bonds per amino acid residue were slightly higher in the proteins of A501 than in those of 3DAC ($p$-value = 2.31e−10, 2-sided Wilcoxon rank sum test), while salt bridges and relative surface areas per amino acid residue performed similarly (salt bridge $p$-value = 0.011 and RSA $p$-value = 0.008, 2-sided Wilcoxon rank sum test). The error bars show the range of distribution. Source data are provided as a Source Data file.

largest portion (>50%) of biomacromolecules[45,46]. Proteome-wide structure predictions were conducted via a high-throughput version of AlphaFold2 (see the "Methods" section). The confidence in the accuracy of the predicted structures can be measured by the predicted local distance difference test (pLDDT) using the lDDT-Cα metric[47]. Residues with pLDDT scores ≥70 are classified as high-confidence residues, which means that >70% of local Cα distances show a predicted error within 4 Å. Residues with pLDDT scores <50 correspond to low confidence[48]. The average pLDDT score of the A501 predictions was 88.63, with 92.8% of proteins (2022 of 2180) having pLDDT scores above 70. For the 3DAC predictions, the average pLDDT score was 82.34, with 79.71% (1194 of 1498) of proteins having pLDDT scores higher than 70. These high pLDDT scores indicate that the structures predicted here are high-confidence structures, showing higher

confidence than is achieved in human proteome predictions (Fig. 2A, Supplementary Fig. 2). In the present work, only structures with pLDDT scores >70 were used for analysis, and the results are presented in the rest of the paper.

We performed statistical analysis of the structural features of all proteins with pLDDT scores >70 in each of the two strains. These features include the commonly used indicators to describe the protein structure, such as protein size (number of amino acids in the protein), secondary structure distribution (α helix, β sheet, and loop), hydrogen bonds, salt bridges, and relative surface area (RSA) (see the "Methods" section). Surprisingly, only the number of hydrogen bonds per amino acid residue was slightly higher in the proteins of A501 than in those of 3DAC ($p$-value = 2.31e−10, 2-sided Wilcoxon rank sum test), while the other indicators performed similarly (salt bridge $p$-value = 0.011 and

RSA $p$-value = 0.008, 2-sided Wilcoxon rank sum test) (Fig. 2, Supplementary Fig. 3). This is in good agreement with the neutron scattering results, that the resilience of the biomolecules in the two species was similar, with the value of A501 being slightly higher (Fig. 1D). In addition, the populations of most amino acids in the proteins were quite similar between A501 and 3DAC (Supplementary Fig. 4). Hence, no significant differences were observed when comparing the average structural features of all proteins between the two strains (Supplementary Note).

## Comparison of protein pairs with the same/similar functions between A501 and 3DAC

In addition to statistical analysis of average protein structural features between the two strains, we also compared the structures of protein pairs with the same or similar function between A501 and 3DAC (see the "Methods" section). A total of 302 genes were identified as orthologs, which is comparable to the proposed minimal gene set (250–300 genes) and less than the estimated genome size for LUCA (500–600 genes)[49]. In addition to the orthologs, protein pairs with the same or similar function but low sequence similarity in A501 and 3DAC were also used for comparison. We classified these protein pairs with the same/similar functions into three groups according to their similarities and differences in sequence, protein structure, and function: (i) orthologous sequences with similar structures and similar functions; (ii) nonorthologous sequences with similar structures and functions; and (iii) nonorthologous sequences with different structures but similar functions (Fig. 1A, Fig. 3, Supplementary Data 1). Here, an orthologous pair is defined as a protein pair with sequence identity above the threshold of 0.3, while a similar structure pair is defined as a protein pair in which the SiMax score between the 3D structures of the two proteins is smaller than 6 Å (see the "Methods" section).

Then, we mapped the above protein pairs to their associated metabolic pathways. To our surprise, a total of 137 protein pairs in the central metabolism and respiratory systems were spontaneously divided into 16 different modules based on the classification of the involved proteins (Fig. 3). In 15 out of the 16 modules, except for the energetic respiratory complex MBH (discussed in the section "Energetic respiratory system"), each metabolic module is dominated by the proteins of a single category. For example, all 19 enzymes in the module of amino acid biosynthesis fall into a group (i), all three enzymes in the pathway of NAD(P)+ biosynthesis belong to group (ii), and all five enzymes in the upstream of glycolysis/gluconeogenesis belong to a group (iii). In addition to these 16 modules, similar categories were also observed in other proteins of biological processes, such as some ribosomal proteins and aminoacyl-tRNA ligase (Fig. 3C, D). In the next two paragraphs, we will discuss the metabolic modules in the central metabolism and respiratory system in detail one by one, since they are closely related to physiological and metabolic functions.

Protein pairs in group (i) are mostly located in the modules associated with central metabolism, including the middle part of the glycolysis/gluconeogenesis pathway between D-glycerate 1,3-diphosphate (G13P2) and acetyl-CoA, in almost the complete purine and pyrimidine biosynthesis pathway, in the biosynthesis and conversion of some amino acids (i.e., Asp, Glu, Ser, Gly, and Thr), and in the biosynthesis of cofactors (i.e., CoA) or precursors (i.e., nicotinate). Group (ii) protein pairs are involved in the link between glycolysis and lipid biosynthesis of G13P2 from dihydroxyacetone phosphate (DHAP) and the biosynthesis of NAD(P)$^+$ from the precursor nicotinate. Notably, these two modules correspond to the pathways in which A501 and 3DAC should be distinct. The module from DHAP to G13P2 is the link to the biosynthesis of lipids, representing the major difference between archaea and bacteria[50,51]. The other is in the biosynthesis of NAD(P)$^+$ from the precursor nicotinate. NAD(P)$^+$ is not stable at high temperatures and can spontaneously degrade at temperatures above 65 °C (the higher the temperature is, the easier the degradation)[52,53]. Group (iii)

protein pairs are involved up- and downstream of glycolysis/gluconeogenesis, in the formation of GAP from glucose (carbohydrate utilization) and production of acetate from acetyl-CoA (acetate production), in the pentose phosphate pathway of the formation of D-ribose 5-phosphate (R5P) from beta-D-fructose 6-phosphate (bF6P), and in the biosynthesis of lipid-associated backbones that are completely different between archaea and bacteria (Fig. 3, Supplementary Fig. 5).

Surprisingly, the energetic respiratory systems in A501 and 3DAC are quite similar, and the vast majority of protein structures are highly conserved between the two, regardless of whether the sequences are similar (Fig. 3). This conserved energetic respiratory system contains a membrane-bound hydrogen-producing complex (MBH) and a membrane-bound sulfane sulfur reduction complex (MBS), as well as soluble hydrogenase I (SH1) and NADP-dependent ferredoxin oxidoreductase II (Nfn2), which maintain the primary redox balance in energetic processes and other metabolic pathways (Fig. 3B)[17]. This respiratory system is considered an ancient system that is a potential ancestor of modern aerobic respiration complex I[54–56], which has only been reported in hyperthermophilic archaea previously[57–60]; this is the first time this system has been reported in deep-branching bacteria. Our results indicate that MBS is more conserved in both sequence and structure than MBH. For MBS, 10 of 13 genes are orthologs with quite similar structures, encoding the complete membrane arm (mbsA, B, C, D, E, G, H, H', and M subunits) and the key catalytic subunit of sulfur reduction (mbsL). The other three proteins (membrane-bound subunit mbsH and the cytosolic subunits mbsK and mbsN) have similar structures, although the similarities of the sequences are slightly lower (identity <30%). In addition, the multimer structure predictions of MBS also showed high structural similarity between strains A501 and 3DAC (Supplementary Fig. 8). Compared to MBS, MBH seems more variable in sequence but still shows relatively high structural conservation (Fig. 3B, E). In MBH, only five genes of 12 are orthologs with similar structures, and the other seven subunits only have similar structures but low sequence identities, except for the membrane-bond subunit mbhB and cytosolic subunit mbhK with SiMax scores of 28 and 12, respectively. Between A501 and 3DAC, the sequence of MBH appears to have more variation in the membrane-bound modules, which may be related to the need to adapt to the completely different structures of the cell membrane in archaea and bacteria. However, the overall highly conserved structure for both MBH and MBS in A501 and 3DAC suggests that this ancient respiratory system can function on the two completely different cell membranes (archaeal and bacterial), consistent with the metabolic observations (Fig. 1D). In addition, the two cytosolic redox balancing complexes, SH1 and Nfn2, which are usually coupled with MBH or/and MBS[61,62], are also highly conserved in both sequence and structure between A501 and 3DAC.

The above results indicate that each module with all the contained proteins evolves as a unit. For example, glycolysis/gluconeogenesis is usually treated as an entire pathway in modern life cells, which would be expected for all proteins in this pathway to change similarly between the two cellular species. However, as seen in Fig. 3A (in the main text), the structures of the proteins in this pathway are clearly classified into groups (i), (ii), and (iii), and these three groups of proteins are separated into distinct modules along the pathway. This finding suggests that all proteins involved in each module may evolve as a unit (Fig. 3A). Metabolic modules of different groups indicated distinct sources and evolutionary histories in modern cells. Both groups (i) and (ii) might dominate the basic functional modules shared by ancient archaea and bacteria, and these are likely conserved metabolic modules acquired from ABCA, linking to the other conserved modules, such as the amino acid biosynthesis module. Compared to the highly conserved group (i), group (ii) may have evolved further independently after the divergence of archaea and bacteria (i.e., the link module between glycolysis and lipid biosynthesis) or may

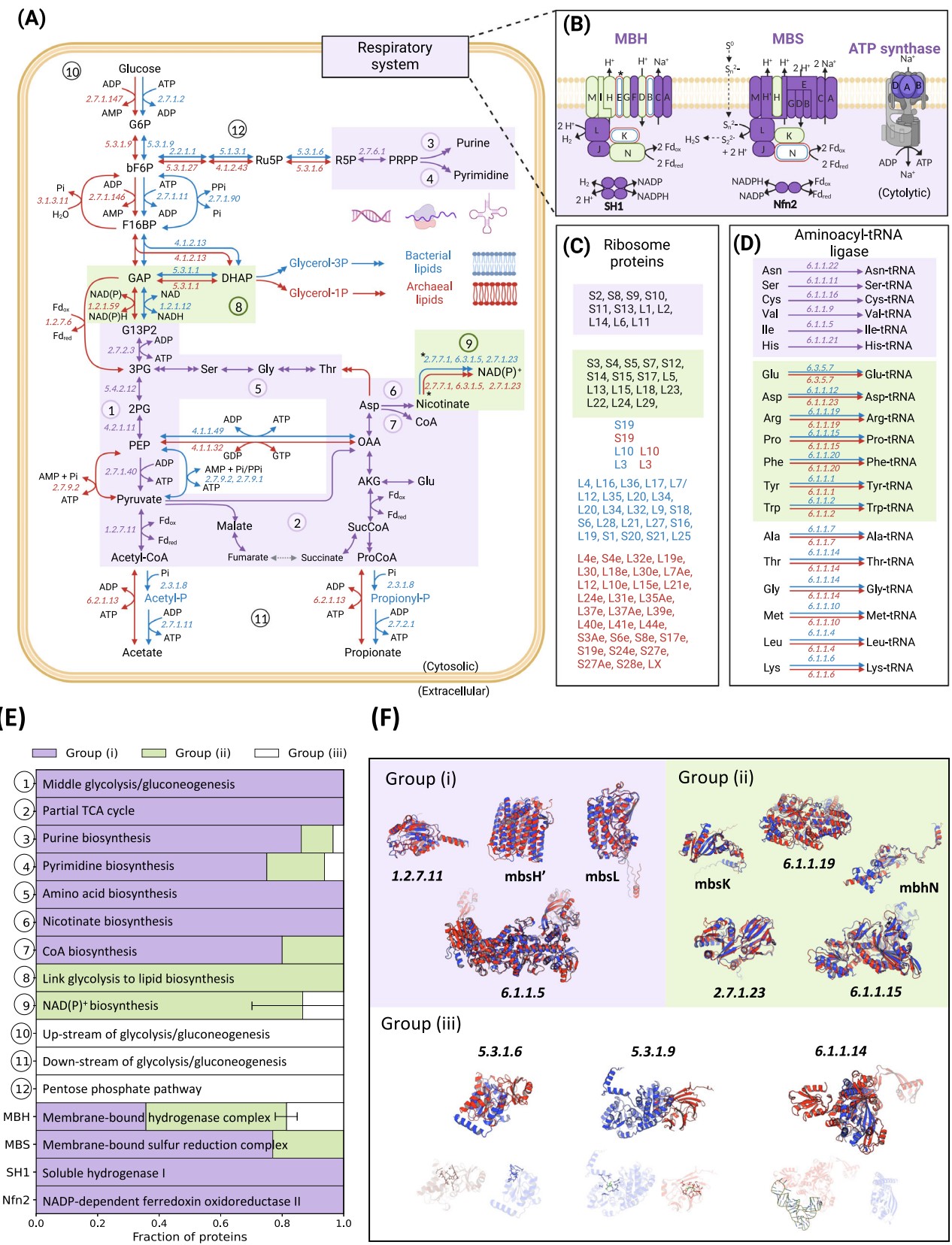

have adapted to different temperatures (i.e., NAD(P)+ biosynthesis). In contrast, the module dominated by group (iii) proteins more likely underwent convergent evolution that occurred after the divergence of archaea and bacteria, and the involved proteins thus have completely different structures and sequences in strains A501 and 3DAC despite performing similar functions. Using the carbon utilization as an example, both A501 and 3DAC are heterotrophs that reasonably lose the oldest $CO_2$ fixation pathway, the Wood–Ljungdahl pathway, but

**Fig. 3 | Metabolic modules identified by classification of involved proteins.**
Classification of proteins is based on the similarities of sequence, structure and function between protein pairs of A501 and 3DAC. Proteins with the same or similar functions were classified into three groups: Group (i), (i) ortholog sequence with similar structure and similar function (in purple); Group (ii) nonortholog sequence but similar structure and function (in green); and Group (iii) nonortholog with different structure but similar function (in white). The cutoff of similar sequences is identity ≥30% (bidirectional best hits with coverage >70% and *p*-value < 1e−5), while the cutoff of similar structures is SiMax score <6 Å. **A** Central metabolism.
**B** Energetic respiratory system. **C** Ribosome proteins. **D** Aminoacyl-tRNA ligase.
**E** Fraction of proteins from three groups in each metabolic module.
**F** Representative cases of protein comparison between A501 (red) and 3DAC (blue) with the same/similar functions in the three groups. Arrows represent biochemical reactions between two compounds. Double arrows indicate that multiple reactions are involved in the conversion between two shown compounds. Numbers in italics beside arrows represent the EC number of enzymes that catalyze the biochemical reactions. Metabolic modules: ① Middle glycolysis/gluconeogenesis (G13P2 to acetyl-CoA); ② Partial TCA cycle; ③ Purine biosynthesis; ④ Pyrimidine biosynthesis; ⑤ Amino acid biosynthesis; ⑥ Nicotinate biosynthesis; ⑦ CoA biosynthesis; ⑧ Link between glycolysis and lipid biosynthesis (DHAP to G13P2); ⑨ NAD(P) + biosynthesis; ⑩ Upstream of glycolysis/gluconeogenesis (glucose to GAP); ⑪ Downstream of glycolysis/gluconeogenesis (acetate/propionate production); ⑫ Pentose phosphate pathway; MBH membrane-bound hydrogenase complex, MBS membrane-bound sulfane sulfur reduction complex; SH1 soluble hydrogenase I; Nfn2 NADP-dependent ferredoxin oxidoreductase II. Abbreviations of metabolites: 2PG D-glycerate 2-phosphate, 3PG 3-phospho-D-glycerate, Acetyl-P acetyl phosphate, AKG 2-oxoglutarate, bF6P beta-D-fructose 6-phosphate, DHAP dihydroxyacetone phosphate, F16BP beta-D-fructose 1,6-bisphosphate, Fd$_{red}$ reduced ferredoxin, Fd$_{ox}$ oxidized ferredoxin, G13P2 D-glycerate 1,3-diphosphate, G13P2 D-glucose 6-phosphate, GAP glyceraldehyde 3-phosphate, Glycerol-3P glycerol 3-phosphate, OAA oxaloacetate, PEP phosphoenolpyruvate, Pi phosphate, PPi diphosphate, ProCoA propionyl-CoA, Propionyl-P propionyl phosphate, R5P D-ribose 5-phosphate, Ru5P D-ribulose 5-phosphate, SucCoA succinate CoA. "*" Represents the two protein pairs, i.e., EC 2.7.7.1 and mbhE, which have the variable normalized RMSD values that shift the group classification using a different model of AlphaFold2. Error bars in **E** represent the variation of group classification in Nicotinate biosynthesis and MBH caused by the variable normalized RMSD values of the above two protein pairs. Source data are provided as a Source Data file.

shift to utilize carbohydrates (e.g., glucose) during the evolution[63]. Even though both A501 and 3DAC can utilize carbohydrates to produce acetate, the function of sugar utilization (the upstream of glycolysis) and acetate production (the downstream of glycolysis) most likely resulted from completely different evolutionary histories, and all in the group (iii). Therefore, we suggested that those structurally conserved proteins in groups (i) and (ii) are more likely to come from a common ancestor that emerged early in the origin of life, and should be more ancient than those in the group (iii) that may come from a late evolutionary addition. These results provide a new view of the underlying evolution and organization of metabolism.

## Extended structural comparison of 12 other bacterial and 12 archaeal representatives

According to the pairwise comparison between A501 and 3DAC, we observed an unexpected scenario in the glycolysis/gluconeogenesis pathway: although it is usually treated as a single complete pathway, our results indicated that it can be composed of four distinct modules (Fig. 3A). The middle two modules show structural conservation between A501 and 3DAC and may have been inherited from the ABCA, while the up- and downstream modules do not show such conservation. To explore the middle modules further, we chose the seven key enzymes catalyzing the reactions of the middle two modules and extended the structure prediction to an additional 12 bacterial and 12 archaeal representatives present in various positions across the phylogenetic tree. These seven key enzymes were glyceraldehyde-3-phosphate dehydrogenase (GAPDH, EC 1.2.1.59/EC 1.2.1.12), triosephosphate isomerase (TIM, EC 5.3.1.1), phosphoglycerate kinase (PGK, EC 2.7.2.3), phosphoglycerate phosphomutase (PGPM, EC 5.4.2.12), enolase (EC 4.2.1.11), pyruvate kinase (PK, EC 2.7.1.40) and pyruvate/2-oxoacid oxidoreductase (POR, EC 1.2.7.11). Among them, five enzymes in the middle part of the glycolysis/gluconeogenesis pathway, namely, PGK, PGPM, enolase, PK, and POR, belong to a group (i), while two enzymes in the link between glycolysis and lipid biosynthesis, namely, GAPDH and TIM, belong to a group (ii). All of the structures of 155 additional proteins were predicted by AlphaFold2 with an average pLDDT score of 94.91, and the pLDDT score of each protein was higher than 70 (Supplementary Data 2).

Groups (i) and (ii), which were identified by comparison between A501 and 3DAC, can be clearly distinguished according to the structural differences among 24 additional archaeal and bacterial species (Fig. 4A), although no clear distinction can be found based on the sequence comparison. The protein structures of five enzymes in group (i) are more highly conserved than those in the two enzymes of the group (ii) according to the comparison of A501, 3DAC, and 24 other archaeal and bacterial strains. For the five enzymes of group (i), their sequences seem too variable and irregular, with the lowest sequence identity of each enzyme being ~20% among 24 additional species, but their structures are highly conserved. This finding confirms the fact that the 3D structure is associated with the function of the proteins more closely than the 1D sequence. For the other two enzymes of the group (ii), both the structures and sequences of GAPDH can be divided into two clusters, namely, EC1.2.1.12 and EC 1.2.1.59, which catalyze the same reaction but with different cofactor specificities. The proteins in the EC 1.2.1.12 cluster are specific to the NAD cofactor, while EC 1.2.1.59[64] can use either NAD or NADP. The conserved active sites in the predicted structures with the PDB templates of experimentally verified enzyme structures supported the distinct enzyme–substrate specificity in the group EC 1.2.1.59 and EC 1.2.1.12 (Fig. 4B, C, see the "Methods" section). The proteins in group EC 1.2.1.59 were all from archaea, while most of the proteins in group EC 1.2.1.12 were from bacteria (Figs. 3, 4A, Supplementary Data 1, Supplementary Figs. 6, 7). A similar status was found for TIM proteins (EC 5.3.1.1). Although the specificity for the substrate is the same, the TIM proteins among different strains can also be divided into two clusters, clusters 1 and 2. The TIM proteins in cluster 2 all belong to archaea, while most TIM proteins in cluster 1 belong to bacteria (Figs. 3, 4B, Supplementary Data 2, Supplementary Figs. 6, 7). The two clusters that were separated primarily based on the structural difference of the above two enzymes (i.e., GAPDH and TIM) are roughly comparable to the two separated branches in each corresponding phylogenetic tree, where proteins in one branch are from archaea, while those in the other are mostly from bacteria (Supplementary Fig. 7A, B). The reactions catalyzed by these two enzymes are directly related to the biosynthesis of glycerol 1-phosphate or glycerol 3-phosphate, which are the divergent precursors for the biosynthesis of archaeal or bacterial lipids, respectively[65–67]. Therefore, these two enzymes still had essential functions in ABCA, and the clear separation of the structures into archaeal and bacterial clusters may be attributed to independent evolution after the divergence of archaea and bacteria with different membrane lipid synthesis precursors (Fig. 3A).

## Discussion

Powerful deep learning tools such as AlphaFold2 make it possible to derive proteome-wide structures from genome sequences with high accuracy[14], which provides more information than using genomics alone and bridges the gap between sequence and function. In this study, we found that protein pairs with similar functions between one representative archaeon (A501) and one representative bacterium (3DAC) could be classified into different groups by judging their commonality and differences in sequence and structure. Interestingly,

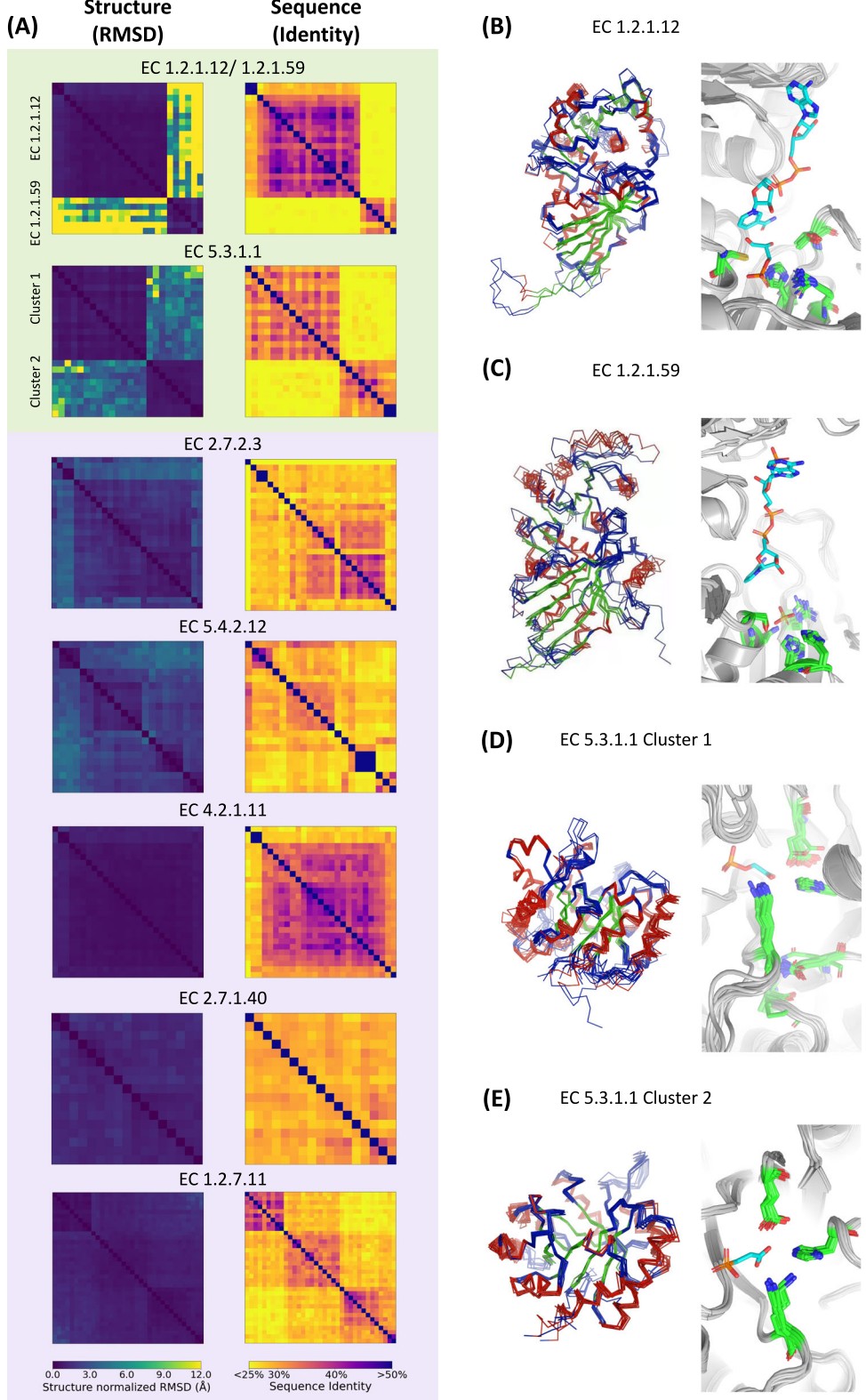

**Fig. 4 | Comparison of seven key enzymes in the glycolysis/gluconeogenesis pathway among A501, 3DAC, and 12 other archaeal and 12 bacterial strains in different positions across the phylogenetic tree. A** Similarity of sequences and structures of seven key enzymes. Superposition predictions of protein structures (left) and reactive sites (right) in the same cluster of proteins with the enzyme classification (EC) number: EC 1.2.1.12 (**B**), EC 1.2.1.59 (**C**) and EC 5.3.1.1 group 1 (**D**) and EC group 2 (**E**). Cluster 1 and Cluster 2 of EC 5.3.1.1. The background colors light green and light purple correspond to the background colors of the pathways shown in Fig. 3A. The PDB IDs of reference protein structure are 1DC4 and 1DC6 for (**B**), 1CF2 for (**C**), 1HG3 for (**D**) and 1AMK for (**E**). Detailed labels and clusters of each heatmap are shown in Supplementary Fig. 6 and Data 2. Source data are provided as a Source Data file.

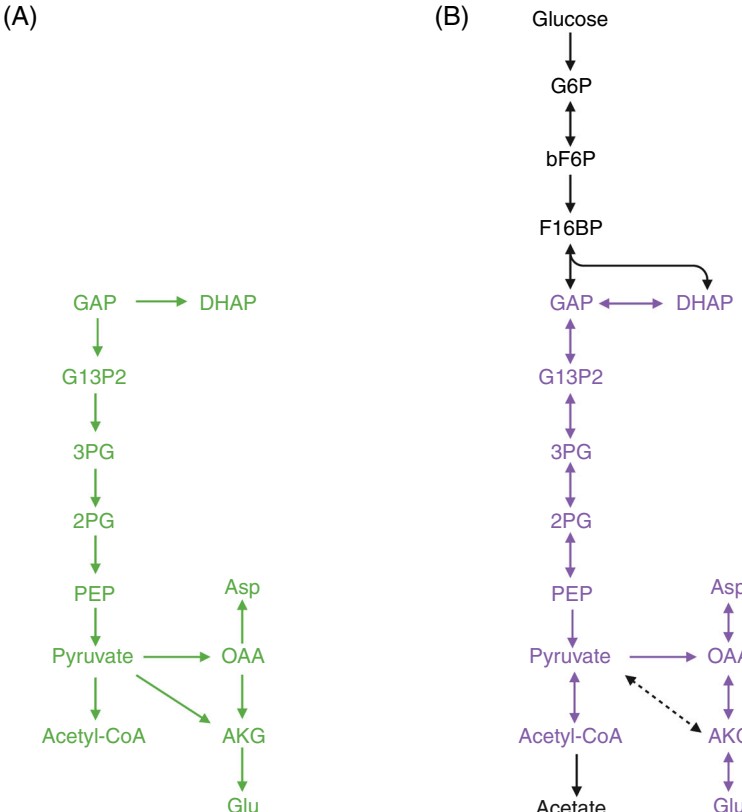

**Fig. 5 | High consistency between the protometabolic pathways in prebiotic chemistry and structural conserved metabolic modules in this study.** Comparison between the proposed protometabolic pathways under 'prebiotically plausible' conditions (**A**) and central carbon metabolism in modern cells (**B**).

Protometabolic pathways were from the chemical evidence in the previous studies[19–29]. Solid arrows in purple and black represent the reactions catalyzed by conserved and variable protein structures in A501 and 3DAC, respectively. The black dashed arrow represents the missing reaction in A501 and 3DAC.

we were able to identify some protein pairs that possess similar structures and functions between the two strains but with low sequence similarity (Supplementary Data 1). This finding further confirms that structure is more closely associated with function than sequence. By mapping the protein pairs of different groups into metabolic pathways, we found that the evolution of proteins in these two strains does not occur at the individual protein level but rather takes place using a metabolic module as a basic unit. Metabolic modules can be used to explore distinct sources and different evolutionary histories. These findings have not been revealed in previous sequence-based analyses.

These new structure-based findings provide new insights into the metabolic organization of ancient archaea, bacteria, and their potential common ancestor (ABCA). Metabolic modules with conserved structures of a series of enzymes shed light on the conserved functions in the ABCA, indicating metabolic modules that likely existed in the ABCA, including the middle half of glycolysis (from DHAP to acetyl-CoA), purine and pyrimidine biosynthesis, metabolism of some essential amino acids (i.e., Asp, Glu, Ser, Gly, and Thr), biosynthesis of some essential cofactors (i.e., NAD(P)$^+$ and CoA), the energetic respiratory system with MBH and MBS, as well as most aminoacyl-tRNA ligases and some ribosome proteins. In particular, for respiratory complexes that are usually hard to identify at the sequence level, our results revealed that ancient MBH and MBS systems can function in both archaea and bacteria at the structural level. In contrast, other metabolic modules with different structures, such as the processes up- (glucose to GAP, for sugar utilization) and

downstream (acetyl-CoA to acetate, for acetate production) of glycolysis/gluconeogenesis and the biosynthesis of lipids, were likely excluded from the metabolism of ABCA. This modular splicing of metabolic combinations in modern cells suggests the possibility that early life may have involved a hybrid of life and nonlife processes, and the connection between metabolic modules may have been abiotic processes.

Notably, the conserved metabolic modules with conserved protein structures between A501 and 3DAC in central carbon metabolism were surprisingly consistent with the experimentally confirmed protometabolic pathways under 'prebiotically plausible' conditions (Fig. 5)[19–29]. The conserved metabolic modules in the group (i) and (ii) that we inferred from the early origin of life are similar to those brought up by prebiotic and evolutionary biochemistry, including the backbone modules of the central metabolism, biosynthesis of essential building blocks, cofactors, as well as energetic and genetic processes[68]. In addition, the metabolic modules dominated by non-conserved protein structures in the group (iii) (e.g., lipid biosynthesis and sugar utilization) that we inferred to obtain during the evolution are in high agreement with the metabolic processes that prebiotic chemistry suggests as a late evolutionary addition[68]. The high overlap between our results and prebiotic chemistry implies that this phenomenon may not be coincidental, but a combination of necessity and chance events intermediated by protein 3D structures under the control of physical and chemical laws. The 3D structure of metabolic enzymes makes a bridge between the sequences in modern cells and the prebiotic chemical reactions, which provides new perspectives to understand

the origin of metabolism. Our results demonstrate the importance of characterizing the 3D structures of proteome-wide protein molecules for understanding the evolutionary mechanism and life origin. We found that the protein structure serves as an important bridge between genomic and metabolic pathways, with the latter having little distinction at the functional level but too much distinction at the sequence level. The comparison of seven key enzymes involved in the glycolysis pathway among 24 archaea and bacterial strains is a good example. Although the sequence similarity among them is low and irregular, the structures are strongly conserved, identifying the essential function that is preserved from ancient life to modern cells. For this reason, we call for extensive use of 3D protein structures, especially proteome-wide structure predictions, as an extension of the sequence-based approach in future research on the origin and evolution of metabolism. Although only a relatively small number of structures have been solved in this study compared to the incredibly diverse prokaryotic proteins, our results proposed that high-throughput protein 3D structure prediction is a promising approach in ancestorial investigations. In the future, if we can combine the vast amount of microbial (meta)genomic information from diverse environments (especially extreme environments similar to early Earth conditions) accumulated over the past decade, high-throughput protein structure prediction could help us summarize the conserved structural and metabolic modules inherited from ancestors in modern cells in a broader context. These structure-based ancestral modules could build a bridge to decipher the key factors that replaced prebiotic native metabolism with enzyme-catalyzed life processes in the origin of life.

## Methods

### Growth and physiological experiments

The specific growth rates under various growth temperatures of strains A501 and 3DAC were measured anaerobically in batch cultures in TRM medium as previously described[15]. Each replicate was set up using a Hungate tube with $N_2$ as the headspace gas in 5 mL of the tested medium. The physiological experiments on amino acid utilization were conducted under optimal conditions of 0.1 MPa for each strain: 85 °C, pH 7.0 and 2.5% NaCl for strain A501, as identified in a previous study;[15] and 70 °C, pH 7.0 and 2.5% NaCl for the strain 3DAC. The TRM basal medium was formulated by removing the carbon sources from the TRM medium. Various single carbon sources were added to a basal TRM medium (2.5% NaCl, containing only 0.002% yeast extract, without any other organic carbon sources) supplemented with 1% (w/v) elemental sulfur. To test carbon source utilization with the addition of 20 kinds of amino acids, various carbon sources at a final concentration of 0.2% (w/v) were added into a basal TRM medium (2.5% NaCl, no yeast extract or tryptone) and supplemented with 20 kinds of amino acids (each at a final concentration of 0.1 g/L) and 1% (w/v) elemental sulfur. To examine the ability of the isolate to grow in the absence of elemental sulfur, cells were cultured in TRM medium (2.5% NaCl, sodium sulfate removed) without sulfur. L-cysteine (20 mM) was tested as the sulfur substitute. For all of the above experiments, resazurin was used as an oxidation–reduction indicator at a final concentration of 0.01% (w/v). All media were adjusted to pH 7.0, autoclaved and then reduced with a drop of 10% (w/v, pH 7.0) $Na_2S$ before inoculation with the cultures of A501 and 3DAC.

### Neutron scattering experiments

A total of 100 mg of cells of A501, 3DAC, and E. coli BL21(DE3) in the late exponential phase were harvested by centrifugation at $6000 \times g$ at 4 °C for 10 min. Cells were transformed into an aluminum sample container for neutron scattering under anaerobic conditions and stored in anaerobic bags.

The neutron scattering experiment of strain A501, 3DAC, and BL21(DE3) was performed on a backscattering EMU spectrometer at ANSTO in Australia with a resolution of $\Delta E = 0.8$ µeV (corresponding to timescales of 1 ns). The elastic scattering intensity $S(q, \Delta t)$ of strains A501, 3DAC, and BL21(DE3) was obtained in the temperature range from 277–310, 277–350, and 277–360 K, respectively, during the heating process at a rate of 1.0 K/min. The scattering vector, $q$, of collected data ranged from 0.27 to 1.97 $\text{Å}^{-1}$. A vanadium sample was measured to define the instrument resolution and to correct for detector efficiency. Intensity data were corrected for sample container and buffer scattering and normalized by vanadium scattering to yield $S(q, \Delta t)$ as a function of temperature for each sample. The sample containers were placed in a nitrogen exchange gas atmosphere to ensure a homogeneous temperature profile. The mean square displacement $\langle x^2(\Delta t) \rangle$ was estimated by using a Gaussian approximation, where $S(q, \triangle t) = \exp(-1/6 \times q^2 \langle x^2(\triangle t) \rangle)$. The effective force constant $\langle k \rangle$ was calculated from $\langle k \rangle = 0.00276/(\text{d} \langle x^2(\triangle t) \rangle / \text{d} T)$[30,31]. The $\langle k \rangle$ errors were calculated from the slope of the weighted straight-line fits to the MSD data by using the Levenberg–Marquardt algorithm.

### Proteome-wide structure prediction

The proteome-wide structure predictions of A501 and 3DAC were constructed by using a modified AlphaFold2 version, called ParaFold[14,69]. In ParaFold, we first built input features including multiple sequence alignment (MSA) and structure templates for all proteins by task parallelism on a supercomputer, followed by parallelized AlphaFold2 inference on a Tesla V100 GPU from NVIDIA DGX-2. Prediction of all A501 and 3DAC protein elapsed totally 143 GPU core hours, which is achievable by using most computer clusters and supercomputers and orders of magnitude faster than the unmodified version of AlphaFold2[48]. In this high-throughput version, we kept the full MSA to preserve the highest accuracy of structural prediction. Predicted structure was refined by AMBER relaxation step since it has a substantial improvement on protein side-chain conformation, which may affect structure properties like hydrogen bonds (Supplementary Figs. 9 and 11). Moreover, we performed all five models that AlphaFold2 provides to predict the structures of each protein in strain A501 and 3DAC. We note that the results using five different models are generally consistent (Supplementary Fig. 2C), and we used the results of all proteins via Model 3 for further analysis. For all structures predicted by the five models, the average pLDDT score of the A501 predictions was 88.63%, with 92.8% of proteins (2022 of 2180) having pLDDT scores above 70 in all 5 models. For the 3DAC predictions, the average pLDDT score was 82.34, with 79.71% (1194 of 1498) of proteins having pLDDT scores higher than 70 in all five models.

### Proteome-wide structure comparison

To conduct statistical analysis of all proteins with pLDDT > 70 between strain A501 and 3DAC, we calculated a series of sequence and structure metrics to characterize their properties. All metrics in our project are calculated through our scripts using PyMOL (The PyMOL Molecular Graphics System, Version 2.0, Schrödinger, LLC.) functions. Hydrogen bonds are defined as polar contacts in proteins, disulfide bonds are calculated by searching two sulfide atoms with distances smaller than 2.1 Å, salt bridges are defined by locating oxygen atoms in acid amino acids, and nitrogen in base amino acid pairs has distances within 4 Å[70]. Relative surface area (RSA) is defined as the quotient of solvent accessible surface area (SASA) and molecular surface area, which is calculated by the PyMOL get_area function. The secondary structure is calculated by the DSS method in PyMOL.

### Protein pair comparison

Initial ortholog mapping between the complete genome sequences of A501 (NCBI accession: CP008887.1)[71] and 3DAC (NCBI accession: CP046447)[17] was performed via a bidirectional best-hit BLAST analysis

as defined in a previous study[72]. The cutoff for ortholog pairs was >30% identity, >70% coverage and an *e*-value < 1e−5 when referenced to each other. The function of each enzyme in the proteomes was annotated by EggNOG[73], KEGG[74], MetaCyc[75], transportDB[76], and TCDB[77] and manually curated according to the literature. The enzyme pairs of strains A501 and 3DAC with the same function were defined as those catalyzing exactly the same reaction, while the enzyme pairs with similar functions were defined as those catalyzing the conversion between at least one substrate and product but with different cofactors. An illustration of metabolic pathways was created with BioRender.com.

The similarity of protein structures between each protein pair was based on the metric called SiMax score. SiMax score is the normalized heavy atoms root mean square deviation (RMSD) between 2 proteins, which was proposed by CATHEDRAL[78] and has been used as a standard in the CATH database[79] to identify 'structure similar groups'. The calculation of the SiMax score is shown in Eq. (1), where $N_{mat}$ represents the number of aligned residues and $L_1$, $L_2$ represent the length of the respective domains. If two proteins in the same or similar functional pair have SiMax < 6 Å, we consider these proteins to have similar structures.

$$\text{SiMax} = \max(L_1, L_2)\frac{\text{RMSD}}{N_{mat}} \qquad (1)$$

### Active site prediction

The structure of the ligand was directly obtained from the co-crystal PDB structure with the ligand embedded inside the active site of the reference protein. The active sites of proteins in Fig. 4 were identified by comparing their structure with those available in the PDB database. First, we picked up several reference protein structures from the PDB database with the same EC number to target proteins with substrates or analogs for the catalytic reaction. The reference protein structures from the PDB database were selected as those with SiMax scores to the target protein smaller than 6 Å. Then, we aligned our predicted protein structures to the reference structures and defined the residues within 4 Å of the substrates in the reference structure as the residues of the active sites. To map the catalytic function and substrate specificity, we compared the active sites of the predicted structure and reference structures. We kept the ligand or analog structure in the reference structure intact. The conserved residue type and similar conformation can support the idea that the target proteins have a similar function as the reference one.

### Phylogenetic and statistical analysis of key enzymes

We also used AlphaFold2 to predict the structures of seven key enzymes in the glycolysis/gluconeogenesis pathway of an additional 12 selected bacterial and 12 archaeal genomes across the tree of prokaryotes. The coding sequences of all 24 genomes were first predicted by Prodigal V2.6.3, and the predicted protein sequences were searched against the locally built reference database of the seven key enzymes in the glycolysis/gluconeogenesis pathway from the KEGG website by Diamond v0.9.24.125 with an *e*-value of 1e−10. Then, those sequences from the 24 genomes were subjected to BLASTP against the KEGG database to confirm their function. In total, the structures of 155 proteins were successfully predicted, with an average pLDDT of 94.91. All the predicted pLDDT values are higher than 70. Based on the predicted structure, we used a heatmap to visualize the SiMax score for each enzyme type and performed hierarchical clustering based on structure similarity (SiMax score) and sequence similarity (sequence identity) by the global alignment function in Biopython[80]. Hierarchical clustering was constructed by using the *cluster.hierarchy* method in the Python SciPy package[81]. The protein sequences of these seven key enzymes from the 24 genomes, as well as those enzymes in A501 and 3DAC, were aligned using the MAFFT algorithm v7.313 with the parameters --ep 0 --genafpair --maxiterate 1000. The phylogenetic tree was built using IQ-Tree v1.6.6 with the model LG + C60 + F + G with a bootstrap value of 1000.

### Reporting summary

Further information on research design is available in the Nature Portfolio Reporting Summary linked to this article.

## Data availability

The complete genome sequences of *Thermococcus eurythermalis* A501 (NCBI accession: CP008887.1) and *Zhurongbacter thermophilus* 3DAC (NCBI accession: CP046447) are available on the NCBI database (https://www.ncbi.nlm.nih.gov/). The proteome-wide structure predictions of *T. eurythermalis* A501 and *Z. thermophilus* 3DAC are available at: https://zenodo.org/record/6300205[82]. Protein structure predictions for 155 additional proteins of 24 additional archaeal and bacterial species are available at: https://zenodo.org/record/6387901[83]. The structure data from other model species are downloaded from AlphaFold database (https://alphafold.ebi.ac.uk/) under a CC-BY-4.0 license. Structures for PDB entries used in this study are available through these links: 1DC4, 1DC6, 1CF2, 1HG3, and 1AMK. Source data are provided with this paper.

## Code availability

The code for processing, analyzing, and visualizing the results is available at: https://github.com/weishuzhao/A501-3DAC-AlphaFold[84]. ParaFold (v1.0) to conduct proteome-wide structure prediction based on AlphaFold (v.2.1.1) pipeline is open-source software (MIT) that is available at: https://github.com/Zuricho/ParallelFold.

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

## Acknowledgements

This study was financially supported by the following funding: the Natural Science Foundation of China (grant numbers 41921006, 42106087, 11974239, and 31630002) for X.X., W.Z., and L.H., the Innovation Program of Shanghai Municipal Education Commission and Shanghai Jiao Tong University Multidisciplinary Research Fund of Medicine and Engineering (project number YG 2016QN13) for L.H., the Oceanic Interdisciplinary Program of Shanghai Jiao Tong University (project number SL2021PT103) for W.Z., and Shanghai Pujiang Program (Grant No. 22PJ1406900) for Z.L. The authors thank the Ph.D. student Aoran Hu from Shanghai Jiao Tong University for help with the pangenome analysis in this work. The authors acknowledge the Center for High-Performance Computing at Shanghai Jiao Tong University for computing resources and the student innovation center at Shanghai Jiao Tong University. The neutron experiment at the EMU of ANSTO was performed under a user program (Proposal No. P9325).

## Author contributions

Conceptualization: X.X., L.H., W.Z., and B.Z.; Research design: W.Z., L.H., X.X., and B.Z.; Proteome-wide structure prediction: B.Z., P.T., and L.H.; Metabolic analysis: W.Z. and X.X.; Data analysis: B.Z., W.Z., L.H., X.X., P.T., and Y.W.; Wet lab experiments: W.Z., H.L., and X.X. Neutron scattering experiment: L.Z., N.S., and Z.L. W.Z., L.H., B.Z., X.X., L.Z., P.T., and Y.W. wrote the manuscript. All authors reviewed the manuscript.

## Competing interests

The authors declare no competing interests.
