## [Peer Review File · Nature Communications]

Proteome-wide 3D structure prediction provides insights into the ancestral metabolism of ancient archaea and bacteriaREVIEWER COMMENTS

Reviewer #1 (Remarks to the Author):

Xiao and co-workers describe an analysis across the 3D structures of the proteome using as modified version of AlphaFold2, called ParaFold. The authors use Parafold to generate atomic models of enzymes from their primary sequence and then compare them. They do this on the proteomes of two representative archaeal and bacterial species, both of which were isolated from deep sea hydrothermal vents. They then use an evaluation of the similarity of the 3D structures for the enzymes associated to a particular metabolic module to sort the various proteins for a particular metabolic function into three groups: (i) related sequence, similar structure (ii) different sequence, but similar structure (iii) different sequence, different structure. Interestingly, when comparing the protein pairs in a particular metabolic pathway, in 15 out of 16 cases it was found that the protein pairs within that pathway fell into the same group. All 19 enzymes related to amino acid biosynthesis fall into group (i), all 5 enzymes in glycolysis/gluconeogenesis fall into group (iii). Interestingly, protein pairs in group (i) are mostly found in central metabolism, including the first part of gluconeogenesis/glycolysis that connects it to the TCA cycle, amino acid biosynthesis, purine and pyrimidine biosynthesis, and the biosynthesis of some cofactors. At least from what I can reason, this would imply they are the oldest. This conclusion is stunning in that it is very similar to what evolutionary biochemists (from phylogenetics and physiology) have been saying for the past two decades and some prebiotic chemists (from simulating metabolic pathways without enzymes) from the past five years, but the authors arrive at the same conclusion from a totally different perspective. Overall, I find this paper to be an important and compelling addition to work on early evolution.

Some questions and comments:

-Is it correct for the reader to interpret that a protein pair found in group (i) should be more ancient than one found in group (iii)? Why or why not? The authors should elaborate on this in the text.

-Line 190-191 "This finding indicates that the enzymes might not have evolved individually but rather evolved by using the metabolic module as a basic unit to adapt to evolutionary pressure." This is a very significant statement, but I think it needs to be elaborated on more. Perhaps additionally restate in a second sentence what is being said in different words. If not, what should be taken away from this? I noticed after writing this that this line is again repeated in line 250-251 and then elaborated on a little, as I had requested above. However, one should consider whether writing the first time is too much all at once. Maybe better to treat it once in lines 250-251?

-One pathway that might be explicitly mentioned and commented on in the text is the Wood-Ljungdahl pathway (aka. Acetyl CoA pathway) for CO₂ fixation. Evolutionary biochemists regularly bring this pathway up as the oldest CO₂ fixation pathway, so what about its enzymes in these two organisms, if they are present? There are also non-enzymatic versions of this pathway as well, such as that reported by Martin [Preiner, M. et al. A hydrogen-dependent geochemical analogue of primordial carbon and energy metabolism. *Nat Ecol Evol* 4, 534–542 (2020).]

-The authors state "Notably, the conserved metabolic modules with conserved protein structures between A501 and 3DAC in central carbon metabolism were surprisingly consistent with the experimentally confirmed protometabolic pathway under "prebiotically plausible" conditions.19-21. I think the references here need to be re-evaluated. The authors cite a paper on the synthesis of PEP by Powner as ref. 19, but in that paper, the authors prepared PEP by a series of chemical steps that do not resemble metabolic pathways, where each step necessitates a step-change in reaction conditions. I think it would be more, or at least equally, relevant to cite earlier papers by Ralser on a non-enzymatic glycolysis [Keller, M. A., Turchyn, A. V. & Ralser, M. Non-enzymatic glycolysis and pentose phosphate pathway-like reactions in a plausible Archean ocean. *Mol Syst Biol* 10, 725 (2014); Keller, M. A. et al. Conditional iron and pH-dependent activity of a non-enzymatic glycolysis and pentose phosphate pathway. *Sci Adv* 2, e1501235 (2016)] and on non-enzymatic gluconeogenesis [Messner, C. B., Driscoll, P. C., Piedrafita, G., Volder, M. F. L. D. & Ralser, M. Nonenzymatic gluconeogenesis-like formation of fructose 1,6-bisphosphate in ice. *Proc National Acad*

Sci 114, 7403–7407 (2017).] These papers actually report the same reactions found in metabolism, not just conceptually similar reactions.

-Likewise, the authors cite two papers on non-enzymatic analogues of the reverse Krebs cycle from Krishnamurthy. Moran published earlier papers on non-enzymatic analogues of the reverse Krebs cycle [Muchowska, K. B. et al. Metals promote sequences of the reverse Krebs cycle. *Nat Ecol Evol* 1, 1716–1721 (2017); Muchowska, K. B., Varma, S. J. & Moran, J. Synthesis and breakdown of universal metabolic precursors promoted by iron. *Nature* 569, 104–107 (2019)]. Considering the hydrothermal nature of the organisms in question and in the conditions used in those papers (especially the 2019 paper), the above paper may be even more, or certainly not less, relevant.

-The authors might also want to consider citing papers on non-enzymatic amino acid synthesis by reductive amination [Barge, L. M., Flores, E., Baum, M. M., VanderVelde, D. G. & Russell, M. J. Redox and pH gradients drive amino acid synthesis in iron oxyhydroxide mineral systems. *Proc National Acad Sci* 116, 201812098 (2019)] and transamination [Mayer, R. J., Kaur, H., Rauscher, S. A. & Moran, J. Mechanistic Insight into Metal Ion-Catalyzed Transamination. *J Am Chem Soc* 143, 19099–19111 (2021)], since the authors also find these to be conserved metabolic modules.

The papers mentioned above further strengthen the author's link between the organisms's environment, their enzymatic similarities and the conditions under which the prebiotic version of the metabolic pathways might have operated.

-It may be of interest to the authors to discuss the overall similarities between the conserved metabolic modules and those often brought up in prebiotic chemistry/evolutionary biochemistry [for example, see Fig. 10 in Muchowska, K. B., Varma, S. J. & Moran, J. Nonenzymatic Metabolic Reactions and Life's Origins. *Chem Rev* 120, 7708–7744 (2020).] As far as I can tell, they are highly similar.

Overall, I very much enjoyed this paper and look forward to it being published.

Reviewer #2 (Remarks to the Author):

Zhao et al. use AlphaFold to predict structures for the proteomes of ancient archaea and bacteria and then cross-reference their results to reveal conserved proteins at the level of protein structure as opposed to protein sequence alone. This approach allows them to identify proteins that have dissimilar sequences but similar structures and functions. Metabolic modules that consist of sets of proteins that have similar evolutionary histories (Groups i, ii, and iii) are also identified. These results are a timely application of current structure prediction tools that, as the authors point out, highlight the importance of considering structure above sequence alone. Though I think most of the claims in the manuscript are justified, I would like the authors to address the following:

(1) The authors note that they omitted the Amber minimization step as it had no influence on "improving model accuracy," where model accuracy appears to have been assessed based on the pLDDT using a C-alpha only pairwise distance order parameter. While it is not surprising that the Amber minimization does not influence the C-alpha trace of predictions much, and therefore does not influence pLDDT greatly, I wonder if it does more to optimize side chains positions (which may influence the hydrogen bond per residue counts in Fig. 2c). Further, I suspect that the minimization may be more important for larger proteins. Have the authors tried using minimized structures to see their influence on the properties calculated in Fig 2? Perhaps the authors can comment on how much the structures tend to change before and after the minimization as a function of protein size?

(2) The authors predicate their analyses downstream of AlphaFold on the first of five models that it produces for each protein, and indicate this seems reasonable because predictions for Model 1 and Model 2 are highly correlated in pLDDT space (Fig. S2c). It would be valuable to the broader community of AlphaFold users to know how much the key results of the manuscript (Fig 3e especially)

results change when different models are used. Further, can the authors use the structures for Models 1, 2, 3, 4, and 5 to estimate confidence intervals for the number of proteins in groups (i), (ii), and (iii) in each module (3e)?

(3) The authors claim (lines 294-295) that superimposed predicted structures for a set of related enzymes with a ligand “confirmed the enzyme substrate specificity.” However, these alignments do not appear to be described in the methods or the figure caption (4B-E), leaving many unanswered questions. Where were the ligand structures obtained? Did the authors perform docking of the ligand into the predicted structures? How do these confirm the specificity? Can the authors quantify how well the ligand fits in the active site for each prediction? Right now, such a strong claim does not seem to be supported by the evidence in Fig 4.

(4) The annotation of function should be more clearly explained. The authors note in the Methods that they use a set of 5 different programs and then manually annotated the results. How many structures had clear literature annotations? For how many do the authors rely on a purely predicted function?

(5) Many thresholds are employed for classifying, for example, orthologous and non-orthologous sequences. How were these thresholds chosen? If they change by +/- 10-20%, how do the results change?

(6) It would be useful to have some information about the proteins that AlphaFold can and cannot make good predictions for in these two proteomes. Can the authors give, e.g., the distributions of protein length and number of domains before and after you apply pLDDT-based filtering to remove poor predictions?

Some additional questions and comments:

(1) I found it confusing that the authors refer to an RMSD metric between A501 and 3DAC structures in the main text (e.g., Fig 3 caption). Many readers will instantly think “how can you take an RMSD between two proteins that have likely diverged in length?” I suggest always using SiMax to refer to this metric to avoid confusion.

(2) Can the authors comment on the prevalence of cis- peptide bonds and incorrect amino acid chirality in predictions produced by AlphaFold?

(3) Line 131, can authors check that values are A501 = 0.31, 3DAC = 0.67, and E. coli = 0.78 N/m? The sentence beginning Line 132 seems to indicate that the actual values should be assigned as A501 = 0.78, 3DAC = 0.67, and E. coli = 0.31, as it says both A501 and 3DAC “showed much higher values than E. coli.”

Reviewer #3 (Remarks to the Author):

The approach followed in this manuscript is in principle quite interesting and may deliver some useful insights. Since my personal expertise lies in the field of scattering techniques I would like to comment only on the neutron scattering part of the manuscript.

(i) I think the given values in the manuscript (see page 6) “The mean resilience values of biomolecules in A501, 3DAC and E. coli were found to be 0.31, 0.67 and 0.78 N/m, respectively” are not correctly assigned. It would understand if 0.31 goes with E. coli??

(ii) In the cited paper from Tehei et al. (ref. 22) the authors also measured E.coli cells and obtained a mean resilience value of 0.39 N/m. Is this deviation in the error limit of the obtained values? What is the error of the k-values obtained in the presented work? The authors from ref. 22 claimed an error of approx. 5 %. That would mean that the value obtained in this paper is out of the error bound. Since the method (determination of mean resilience values from different cell types) was to my knowledge not applied in many studies, it is methodically unclear for me how reliable and meaningful the determined values are. Is the difference in k-values between A501 and 3DAC statistically significant?

The authors should give more details on the methodically aspects of this approach in order to convince the reader, that the use neutron scattering gives trustable results.

(iii) The authors should give information on the q-range of the recorded data.

Reviewer's Comments:

(point-by-point)

Reviewer #1 (Remarks to the Author)

Xiao and co-workers describe an analysis across the 3D structures of the proteome using as modified version of AlphaFold2, called ParaFold. The authors use Parafold to generate atomic models of enzymes from their primary sequence and then compare them. They do this on the proteomes of two representative archaeal and bacterial species, both of which were isolated from deep sea hydrothermal vents. They then use an evaluation of the similarity of the 3D structures for the enzymes associated to a particular metabolic module to sort the various proteins for a particular metabolic function into three groups: (i) related sequence, similar structure (ii) different sequence, but similar structure (iii) different sequence, different structure. Interestingly, when comparing the protein pairs in a particular metabolic pathway, in 15 out of 16 cases it was found that the protein pairs within that pathway fell into the same group. All 19 enzymes related to amino acid biosynthesis fall into group (i), all 5 enzymes in glycolysis/gluconeogenesis fall into group (iii). Interestingly, protein pairs in group (i) are mostly found in central metabolism, including the first part of gluconeogenesis/glycolysis that connects it to the TCA cycle, amino acid biosynthesis, purine and pyrimidine biosynthesis, and the biosynthesis of some cofactors. At least from what I can reason, this would imply they are the oldest. This conclusion is stunning in that it is very similar to what evolutionary biochemists (from phylogenetics and physiology) have been saying for the past two decades and some prebiotic chemists (from simulating metabolic pathways without enzymes) from the past five years, but the authors arrive at the same conclusion from a totally different perspective. Overall, I find this paper to be an important and compelling addition to work on early evolution.

Response: We appreciate the comments and suggestions from the reviewer for the deep understanding, encouragement and recognition of the significance of our work. According to these comments, we have further discussed the significance of our results with the overall similarities from prebiotic chemistry and added additional evidence to support these conclusions based on suggested references. Line-by-line response is as following:

-Is it correct for the reader to interpret that a protein pair found in group (i) should be more ancient than one found in group (iii)? Why or why not? The authors should elaborate on this in the text.

Response: Yes, we agreed that a protein pair found in group (i) should be more ancient than one found in group (iii) as the reviewer mentioned. To make the statement clearer and easier to understand, we rephrased the potential reasons with detailed examples, the content as following: “Both groups (i) and (ii) might dominate the basic functional modules shared by ancient archaea and bacteria, and these are likely conserved metabolic modules acquired from ABCA, linking to the other conserved modules, such as the amino acid biosynthesis module. Compared to the highly conserved group (i), group (ii) may have evolved further independently after the divergence of archaea and bacteria (i.e., the link module between glycolysis and lipid biosynthesis) or may have adapted to different temperatures (i.e., NAD(P)⁺ biosynthesis). In contrast, the module dominated by group (iii) proteins more likely underwent convergent evolution that occurred after the divergence of archaea and bacteria, and the involved proteins thus have completely different structures and sequences in strains A501 and 3DAC despite performing similar functions. Using the carbon utilization as example, both A501 and 3DAC are heterotrophs that reasonably lose the oldest CO₂ fixation pathway, Wood-Ljungdahl pathway, but shift to utilize the carbohydrates (e.g., glucose) during the evolution⁵⁵. Even though both A501 and 3DAC can utilize carbohydrates to produce acetate, the function of sugar utilization (the upstream of glycolysis) and acetate production (the downstream of glycolysis) are most likely resulted from completely different evolutionary histories, and all in group (iii). Therefore, we suggested that those structurally conserved proteins in groups (i) and (ii) are more likely to come from a common ancestor that emerged early in the origin of life, and should be more ancient than those in group (iii) that may come from a late evolutionary addition.” We have added the above discussions into the revised manuscript in line 252-273 in page 12-13.

-Line 190-191 “This finding indicates that the enzymes might not have evolved individually but rather evolved by using the metabolic module as a basic unit to adapt to evolutionary pressure.” This is a very significant statement, but I think it needs to be elaborated on more. Perhaps additionally restate in a second sentence what is being said in different words. If not, what should be taken away from this? I noticed after writing this that this line is again repeated in line 250-

251 and then elaborated on a little, as I had requested above. However, one should consider whether writing the first time is too much all at once. Maybe better to treat it once in lines 250-251?

Response: We thank the careful reading and helpful suggestion from the reviewer. We agreed that a thorough elaboration of this statement at once is a better choice. To make the statement more accurate, we have modified our statement in the revised manuscript from “The above results indicate that the enzymes might not have evolved individually but rather takes place using a metabolic module as a basic unit to adapt to evolutionary pressure” to “The above results indicate that each module with all the contained proteins evolve as a unit.”. We further explained it with the following sentences: “For example, glycolysis/gluconeogenesis is usually treated as an entire pathway in modern life cells, which would be expected all proteins in this pathway changed similarly between the two cellular species. However, as seen in Fig. 3A (in the main text), the structures of the proteins in this pathway are clearly classified into group (i), (ii) and (iii), and these three groups of proteins are separated into distinct modules along the pathway. This finding suggests that all proteins involved in each module may evolve as a unit.” We have added the above discussion in the revised manuscript in line 245 to 251 in page 12.

-One pathway that might be explicitly mentioned and commented on in the text is the Wood-Ljungdahl pathway (aka. Acetyl CoA pathway) for CO₂ fixation. Evolutionary biochemists regularly bring this pathway up as the oldest CO₂ fixation pathway, so what about its enzymes in these two organisms, if they are present? There are also non-enzymatic versions of this pathway as well, such as that reported by Martin [Preiner, M. et al. A hydrogen-dependent geochemical analogue of primordial carbon and energy metabolism. *Nat Ecol Evol* 4, 534–542 (2020).]

Response: We appreciate the reviewer for pointing out the Wood-Ljungdahl (WL) pathway. Indeed, the WL pathway is usually treated as the oldest CO₂ fixation pathway as the reviewer mentioned. However, the archaeal and bacterial strains we used in this study are both obligate heterotrophs which were commonly thought to lose the CO₂ fixation pathways but shift to using organic carbon sources (e.g., glucose) during the evolution. Our results indicate that the glucose utilization enzymes in glycolysis pathway were in group (iii), differ in both sequence and structure, may support the assumption that the organic carbon utilization pathways were not from the origin of life but from a late evolutionary addition.

We accepted the reviewer's suggestion and mentioned the missing of WL pathway and the late evolutionary addition of glucose utilization in the revised manuscript in line 261 to 274 in page 12-13, with the following content: “Using the carbon utilization as example, both A501 and 3DAC are heterotrophs that reasonably lose the oldest CO₂ fixation pathway, Wood-Ljungdahl pathway, but shift to utilize the carbohydrates (e.g., glucose) during the evolution⁵⁵. Even though both A501 and 3DAC can utilize carbohydrates to produce acetate, the function of sugar utilization (the upstream of glycolysis) and acetate production (the downstream of glycolysis) are most likely resulted from completely different evolutionary histories, and all in group (iii). Therefore, we suggested that those structurally conserved proteins in groups (i) and (ii) are more likely to come from a common ancestor that emerged early in the origin of life, and should be more ancient than those in group (iii) that may come from a late evolutionary addition.” Here we cited the reference for WL pathway suggested by the reviewer.

-The authors state “Notably, the conserved metabolic modules with conserved protein structures between A501 and 3DAC in central carbon metabolism were surprisingly consistent with the experimentally confirmed protometabolic pathway under “prebiotically plausible” conditions.19-21. I think the references here need to be re-evaluated. The authors cite a paper on the synthesis of PEP by Powner as ref. 19, but in that paper, the authors prepared PEP by a series of chemical steps that do not resemble metabolic pathways, where each step necessitates a step-change in reaction conditions. I think it would be more, or at least equally, relevant to cite earlier papers by Ralser on a non-enzymatic glycolysis [Keller, M. A., Turchyn, A. V. & Ralser, M. Non-enzymatic glycolysis and pentose phosphate pathway-like reactions in a plausible Archean ocean. *Mol Syst Biol* 10, 725 (2014); Keller, M. A. et al. Conditional iron and pH-dependent activity of a non-enzymatic glycolysis and pentose phosphate pathway. *Sci Adv* 2, e1501235 (2016)] and on non-enzymatic gluconeogenesis [Messner, C. B., Driscoll, P. C., Piedrafita, G., Volder, M. F. L. D. & Ralser, M. Nonenzymatic gluconeogenesis-like formation of fructose 1,6-bisphosphate in ice. *Proc National Acad Sci* 114, 7403–7407 (2017).] These papers actually report the same reactions found in metabolism, not just conceptually similar reactions.

Response: We are very grateful to the reviewer for the professional suggestion and detailed information. In the revised manuscript, we already added the reference following the reviewer’s

suggestion, numbered as [60] and [61], in line 358-361 in page 17. These references were also cited in the legend of Fig. 5.

-Likewise, the authors cite two papers on non-enzymatic analogues of the reverse Krebs cycle from Krishnamurthy. Moran published earlier papers on non-enzymatic analogues of the reverse Krebs cycle [Muchowska, K. B. et al. Metals promote sequences of the reverse Krebs cycle. *Nat Ecol Evol* 1, 1716–1721 (2017); Muchowska, K. B., Varma, S. J. & Moran, J. Synthesis and breakdown of universal metabolic precursors promoted by iron. *Nature* 569, 104–107 (2019)]. Considering the hydrothermal nature of the organisms in question and in the conditions used in those papers (especially the 2019 paper), the above paper may be even more, or certainly not less, relevant.

Response: We thank the professional suggestion and detailed information from the reviewer again. In the revised manuscript, we already added the reference following the reviewer's suggestion, numbered [62] and [63], in line 358-361 in page 17. These references were also cited in the legend of Fig. 5.

-The authors might also want to consider citing papers on non-enzymatic amino acid synthesis by reductive amination [Barge, L. M., Flores, E., Baum, M. M., VanderVelde, D. G. & Russell, M. J. Redox and pH gradients drive amino acid synthesis in iron oxyhydroxide mineral systems. *Proc National Acad Sci* 116, 201812098 (2019)] and transamination [Mayer, R. J., Kaur, H., Rauscher, S. A. & Moran, J. Mechanistic Insight into Metal Ion-Catalyzed Transamination. *J Am Chem Soc* 143, 19099–19111 (2021)], since the authors also find these to be conserved metabolic modules.

The papers mentioned above further strengthen the author's link between the organisms's environment, their enzymatic similarities and the conditions under which the prebiotic version of the metabolic pathways might have operated.

Response: We are very happy to know the non-enzymatic amino acids synthesis also support our observation. In the revised manuscript, we already added the reference following the reviewer's suggestion, numbered as [64] and [65], in line 358-361 in page 17. These references were also cited in the legend of Fig. 5.

-It may be of interest to the authors to discuss the overall similarities between the conserved metabolic modules and those often brought up in prebiotic chemistry/evolutionary biochemistry [for example, see Fig. 10 in Muchowska, K. B., Varma, S. J. & Moran, J. Nonenzymatic Metabolic Reactions and Life's Origins. Chem Rev 120, 7708–7744 (2020).] As far as I can tell, they are highly similar.

Response: We thank this very constructive suggestion from the reviewer, and we learned a lot of from the excellent review provided by the reviewer. According to reviewer's suggestion, we expanded the discussion of the overall similarities between the conserved metabolic modules and those often brought up in prebiotic chemistry/evolutionary biochemistry as a separated paragraph in the Discussion section, in line 358-373 in page 16-17, with the citation of suggested reference numbered as [66] in revised manuscript.

Here is the content of this paragraph: “Notably, the conserved metabolic modules with conserved protein structures between A501 and 3DAC in central carbon metabolism were surprisingly consistent with the experimentally confirmed protometabolic pathways under ‘prebiotically plausible’ conditions (**Fig. 5**)^{19-21,60-65}. The conserved metabolic modules in the group (i) and (ii) that we inferred from the early origin of life are similar to those brought up by prebiotic and evolutionary biochemistry, including the backbone modules of the central metabolism, biosynthesis of essential building blocks, cofactors, as well as energetic and genetic processes⁶⁶. In addition, the metabolic modules dominated by non-conserved protein structures in group (iii) (e.g., lipid biosynthesis and sugar utilization) that we inferred to obtain during the evolution are in high agreement with the metabolic processes that prebiotic chemistry suggests as a late evolutionary addition⁶⁶. The high overlap between our results and prebiotic chemistry implies that this phenomenon may not be coincidental, but a combination of necessity and chance events intermediated by protein 3D structures under the control of physical and chemical laws. The 3D structure of metabolic enzymes makes a bridge between the sequences in modern cells and the prebiotic chemical reactions, which provides new perspectives to understand the origin of metabolism.”

Overall, I very much enjoyed this paper and look forward to it being published.

Reviewer #2 (Remarks to the Author)

Zhao et al. use AlphaFold to predict structures for the proteomes of ancient archaea and bacteria and then cross-reference their results to reveal conserved proteins at the level of protein structure as opposed to protein sequence alone. This approach allows them to identify proteins that have dissimilar sequences but similar structures and functions. Metabolic modules that consist of sets of proteins that have similar evolutionary histories (Groups i, ii, and iii) are also identified. These results are a timely application of current structure prediction tools that, as the authors point out, highlight the importance of considering structure above sequence alone. Though I think most of the claims in the manuscript are justified, I would like the authors to address the following:

Response. We thank the reviewer's recognition that this work is a timely application of current structure prediction tools, and of course, this kind of application is still in its infancy and figuring out a standardized procedure to follow. Therefore, we are very grateful for the reviewer's constructive suggestions that helped us improve the technical persuasiveness of this manuscript. Line-by-line response is as following:

(1) The authors note that they omitted the Amber minimization step as it had no influence on "improving model accuracy," where model accuracy appears to have been assessed based on the pLDDT using a C-alpha only pairwise distance order parameter. While it is not surprising that the Amber minimization does not influence the C-alpha trace of predictions much, and therefore does not influence pLDDT greatly, I wonder if it does more to optimize side chains positions (which may influence the hydrogen bond per residue counts in Fig. 2c). Further, I suspect that the minimization may be more important for larger proteins. Have the authors tried using minimized structures to see their influence on the properties calculated in Fig 2? Perhaps the authors can comment on how much the structures tend to change before and after the minimization as a function of protein size?

Response: We thank the advises from the reviewer. We agreed that the relax step in AlphaFold is essential for side chain conformations. To answer the reviewer's question, we recalculated all protein structures from A501 and 3DAC proteome with all five AlphaFold models with relax step, the Amber minimization. We found the reviewer is correct, and the relax step significantly influenced AlphaFold predicted models in their structure properties, so we agree that the relax

step is necessary for structure analysis. Therefore, we have recalculated the features used in our paper (including hydrogen bonds, salt bridges, disulfide bonds, and RSA), and compared these features before and after relax step. As seen in Fig. R1 below, which is Fig. S9 in the revised SI, after the relax step, the hydrogen bond per residue, salt bridge per residue and disulfide bonds has increased, and RSA has decreased. So, we rerun all our analysis results based on our relaxed structures, and select the best performance model 3 for our analysis. This modification affects Fig. 2, and Fig. S3, supplemental datasets and statistical results in our manuscripts (the comparison of previous and updated figures were attached below). Although the Amber minimization has quantitative impacts on the predicted protein structures (i.e., hydrogen bonds, salt-bridges, and relative surface areas), it will not change the relevant conclusions presented in the previous manuscript. The updated results still agreed with the conclusion of proteome-wide structure comparison that the salt bridges and RSA are generally similar between the A501 and 3DAC. Still, hydrogen bonds are significantly slightly higher in A501 than in 3DAC (see the comparison in Fig. R2 attached below). More importantly, the metabolic modules differentiated according to structural similarity are not affected by whether a relax step is performed or not, and the results obtained are almost the same to the previous manuscript (see the comparison in Fig. R3 attached below).

To describe the above processes, we added a paragraph in the “Proteome-wide structure prediction” section in revised manuscript, with the content as: “Predicted structure was refined by AMBER relaxation step, since it has substantial improvement on protein side chain conformation, which may affect structure properties like hydrogen bonds (Fig. S9 and S11). Moreover, we performed all five models that AlphaFold2 provides to predict structures of each protein in strain A501 and 3DAC. We note that the results using five different models are generally consistent (**Supplemental Fig. S2C**), and we used the results of all proteins via Model 3 for further analysis. For all structures predicted by five models, the average pLDDT score of the A501 predictions was 88.63, with 92.8% of proteins (2,022 of 2,180) having pLDDT scores above 70 in all 5 models. For the 3DAC predictions, the average pLDDT score was 82.34, with 79.71% (1,194 of 1,498) of proteins having pLDDT scores higher than 70 in all five models.”, shown in line 450 to 459 in page 21.

Fig. R1. (Supplemental Fig. S9.) differences in Structural properties before and after the AlphaFold relax step. The A501 protein structures are colored in red, while 3DAC structures are represented in blue. The comparison of structural properties by subtracting the values of the unrelaxed structures from those of the corresponding relaxed ones, for hydrogen bond per residue **(A)**, salt bridge per residue **(B)**, RSA **(C)** and disulfide bond **(D)**. After relax, interaction between residues like hydrogen bond, salt bridge and disulfide bond has increased, while RSA has decreased. This supports that the relax step in AlphaFold is crucial for structure analysis.

(A) Difference in hydrogen bond per residue

(B) Difference in salt bridge per residue

(C) Difference in RSA (relative surface area)

(D) Difference in disulfide bond

(R2-A) old version without relax step

(R2-B) updated version with relax step

Fig. R2 (Fig. 2 in the main text). Construction of proteome-wide protein structures and statistical comparison between A501 (red) and 3DAC (blue). (A) Composition of average pLDDT per residue in proteome-wide protein structures in A501 and 3DAC compared with the model organisms. (B) Comparison of helix-sheet percentage between A501 and 3DAC. (C) Comparison of hydrogen bonds, salt bridges and relative surface areas per amino acid residue in all proteins of A501 and 3DAC.

(R3-A) old version without relax step

(R3-B) updated version with relax step

Fig. R3. (Fig. 3 in the main text), “*” represents the two protein pairs, i.e., EC 2.7.7.1 and mbhE, have the variable normalized RMSD values that shift the group classification using different model of AlphaFold2. Error bars in Fig. 3E represent the variation of group classification in Nicotinate biosynthesis and MBH caused by the variable normalized RMSD values of above two protein pairs. (This information has been added into the updated legend of Fig. 3)

(2) The authors predicate their analyses downstream of AlphaFold on the first of five models that it produces for each protein, and indicate this seems reasonable because predictions for Model 1 and Model 2 are highly correlated in pLDDT space (Fig. S2c). It would be valuable to the broader community of AlphaFold users to know how much the key results of the manuscript (Fig 3e especially) results change when different models are used. Further, can the authors use the structures for Models 1, 2, 3, 4, and 5 to estimate confidence intervals for the number of proteins in groups (i), (ii), and (iii) in each module (3e)?

Response: According to reviewer's question, we have predicted AlphaFold structures for proteome-wide proteins of A501 and 3DAC strains with all five models as we mentioned above. We found the pLDDT between all five models are still consistent (see the detailed comparison in Fig. R4 attached below, which is numbered as Supplemental Fig. S2 in updated SI).

Based on our predicted structures from five models, we calculated SiMax (RMSD) for all protein pairs, showing in the updated supplemental dataset S1. Based on our new results, we found most of the classification of group (i), (ii) and (iii) keep consistent in predictions of all five models. Only two proteins varied with different models in Fig. 3 are marked by "*" in the revised version (see the comparison in Fig. R3 mentioned above). The only two proteins, EC 2.7.7.1 and mbhE, which have inconsistent classification results in 5 models, impacted the protein counts in group (ii) or group (iii) in two metabolic modules, Nicotinate biosynthesis and MBH in the Fig. 3E. The confidence intervals of the classifications of these two metabolic modules are now shown in revised Fig. 3E (see the detailed comparison in Fig. R5 attached below).

Fig. R4 (Supplemental Fig. S2.) Proteome-wide structure predictions of A501 and 3DAC. **(A)** Composition protein structure predictions with average pLDDT per protein in A501 and 3DAC, compared with the model organisms. **(B-C)** pLDDT score present high consistency in fraction and high correlation when using different models (compare model 3 to all other models) in AlphaFold2-based pipeline **(D)**.

(A) pLDDT per protein in A501 and 3DAC results (model 3) and other species from AlphaFold database

(B) pLDDT per protein of different AlphaFold models in A501 and 3DAC results

(C) pLDDT per residue of different AlphaFold models in A501 and 3DAC results

(D) Correlation of average protein pLDDT of different AlphaFold models

(R5-A) old version without relax step

(R5-B) updated version with relax step

Fig. R5 (Fig. 3E in the main text). Error bars in Fig. 3E represent the variation of group classification in Nicotinate biosynthesis and MBH caused by the variable normalized RMDS values of above two protein pairs. (This information has been added into the updated legend of Fig. 3)

(3) The authors claim (lines 294-295) that superimposed predicted structures for a set of related enzymes with a ligand “confirmed the enzyme substrate specificity.” However, these alignments do not appear to be described in the methods or the figure caption (4B-E), leaving many unanswered questions. Where were the ligand structures obtained? Did the authors perform docking of the ligand into the predicted structures? How do these confirm the specificity? Can the authors quantify how well the ligand fits in the active site for each prediction? Right now, such a strong claim does not seem to be supported by the evidence in Fig 4.

Response: We thank the reviewer for careful reading. We agreed that the claim is too strong and may be misunderstood. According to the question from the reviewer, we rephrased the statement in the results as: “The conserved active sites in the predicted structures with the PDB references of experimentally verified enzyme structures supported the distinct enzyme-substrate specificity in the group EC 1.2.1.59 and EC 1.2.1.12 (Fig. 4B-C, Materials and Methods).” in line 308-310 in page 14 in the revised manuscript. In addition, the PDB reference structure IDs used for EC

1.2.1.59 and EC 1.2.1.12 in Fig. 4B-D were also added into the legend of Fig. 4, with the content as: “The PDB IDs of reference protein structure are 1DC4 and 1DC6 for (B), 1CF2 for (C), 1HG3 for (D) and 1AMK for (E)”.

At the same time, to describe the methods clearer, we rephrased the content in “Active site prediction” section in Materials and Methods in line 526-537 in page 24 in revised manuscript, with the following content: “The structure of the ligand was directly obtained from the co-crystal PDB structure with the ligand embedded inside the active site of the reference protein. The active sites of proteins in **Fig. 4** were identified by comparing their structure with those available in the PDB database. First, we picked up several reference protein structures from the PDB database with the same EC number to target proteins with substrates or analogs for the catalytic reaction. The reference protein structures from the PDB database were selected as those with *SiMax* score to the target protein smaller than 6 Å. Then, we aligned our predicted protein structures to the reference structures and defined the residues within 4 Å of the substrates in the reference structure as the residues of the active sites. To map the catalytic function and substrate specificity, we compared the active sites of the predicted structure and reference structures. We kept the ligand or analog structure in the reference structure intact. The conserved residue type and similar conformation can support the idea that the target proteins have a similar function as the reference one.” So, we did not perform docking of the ligand, and specificity is confirmed by the similar conformation and residue conservation of the active site.

(4) The annotation of function should be more clearly explained. The authors note in the Methods that they use a set of 5 different programs and then manually annotated the results. How many structures had clear literature annotations? For how many do the authors rely on a purely predicted function?

Response: We thank the careful reading from the reviewer. We used these different annotation tools depending on their different specialties to obtain as much as possible potential functional prediction. For example, EggNOG is general annotation tool, the transportDB and TCDB are specialized on the collection of transmembrane proteins, whereas KEGG and MetaCyc is better at annotating intracellular reactions. Then we use the manual curation according to the literature reported biochemistry and physiological experiments to verify the annotation from these automatic approaches on public databases. For protein structures, 64% of A501 proteins have

PDB supporting ortholog, and 61% for 3DAC proteins. For the functional annotation, in this study, we focus on the two major functional categories, i.e., the Biosynthesis and metabolism, and the Energy production and conversion. Among 201 protein pairs (both ortholog and non-ortholog) of A501 and 3DAC in these two categories, literature supporting annotations of protein pairs are 91 in total, covered almost all enzymes in the central carbon metabolism and respiration processes shown in Fig. 3A-B. There are only 40 protein pairs in these two functional categories were rely on predicted functions from the public database without either literature or PDB supporting. We added the reference IDs for the literature supporting protein pairs and PDB IDs of the PDB orthologs in the Supplemental Dataset S1 to make it clearer. Above discussion has been added into the revised SI text under the “Detailed description of the functional annotation” section.

(5) Many thresholds are employed for classifying, for example, orthologous and non-orthologous sequences. How were these thresholds chosen? If they change by +/- 10-20%, how do the results change?

Response: We thank the careful check from the reviewer. In this study, we used a commonly accepted threshold to identify the ortholog group with the identity $\geq 30\%$, coverage $\geq 70\%$ of bidirectional mapping, and e-value $< 1e-3$ as described in previous study (we cited in the manuscript) [Ref: <https://doi.org/10.3389/fmicb.2014.00110>]. All protein pairs analyzed with functions and classified into group (i) and (ii) and (iii) are shown in Fig. R2 (added as Supplemental Fig. S13 in revised SI). It is obvious that the threshold for sequence cannot distinct the protein pairs with similar structures or not, which supports our statement that structure can tell more than sequence in the main text. In addition, the main statement of the metabolic modules won't be changed no matter which threshold we chose for the sequence, because our conclusions mainly depend on the similarity of structures and the sequence similarity are only use as a comparison. If we shift coverage threshold +/- 10-20%, it only changed the classification of a few protein pairs within group (i) and (ii) (see Fig. R6 below). For our statement, both group (i) and (ii) are conserved and most likely from the common ancestor of archaea and bacteria, and the group (ii) is more variable in sequence than group (i). So, the main conclusion won't be changed. For the sequence identity, the current threshold of 30% identity is the commonly used to identity orthologs, and it is believed to be associated with the evolution time: the change from

30% to 20% might correspond to a billion-year divergence time change [Ref: doi: 10.1002/0471250953.bi0301s42]. Even if we shift the identity threshold +/- 10-20%, the different metabolic modules can also be identified based on the structure similarity. The above discussion and Fig. R6 have been added into the revised SI as Supplemental Fig. S13, and above discussions were also included into the “Discussion about the threshold of ortholog groups used in this study” section in the revised SI text.

Figure R6. (Supplemental Fig. S13) Distribution of protein pairs analyzed with functions and classified into group (i) and (ii) and (iii) in this study. The dashed lines represent the threshold used in this study.

(6) It would be useful to have some information about the proteins that AlphaFold can and cannot make good predictions for in these two proteomes. Can the authors give, e.g., the distributions of protein length and number of domains before and after you apply pLDDT-based filtering to remove poor predictions?

Response: We thank the reviewer’s suggestion. To investigate the differences between high and

low-pLDDT proteins, we did further analysis on correlations of pLDDT to some protein properties. We added a supplemental Fig. S10 (see Fig. R7 attached below) to show the correlation between protein average pLDDT to protein length and protein loop percentage. We also present the distribution of protein length between all proteome and pLDDT-filtered proteome, and find the difference in protein length distribution before and after pLDDT filter is not significant by Wilcoxon ranked sum test (A501 p-value: 0.05307, 3DAC p-value: 0.7248).

Fig. R7 (Supplemental Fig. S10.) Structural analysis between high and low-pLDDT proteins. (A) Compared the correlation between protein length and protein average pLDDT. (B) Compared the correlation between protein loop percentage and protein average pLDDT. (C) presented the distribution of protein length before and after pLDDT-based filter (all prediction models' pLDDT > 70). The Wilcoxon rank sum test between the distribution showed there is no significant difference in protein length (A501 p-value: 0.05307, 3DAC p-value: 0.7248)

(A) Correlation between protein length and protein average pLDDT

(B) Correlation between protein loop percentage and protein average pLDDT

(C) Protein length distribution differences before and after pLDDT-filter

Some additional questions and comments:

(1) I found it confusing that the authors refer to an RMSD metric between A501 and 3DAC structures in the main text (e.g., Fig 3 caption). Many readers will instantly think “how can you take an RMSD between two proteins that have likely diverged in length?” I suggest always using *SiMax* to refer to this metric to avoid confusion.

Response: We thank the suggestion from the reviewer. We changed all of the "RMSD" term to "*SiMax*" in our manuscript to avoid this confusion. The sentences used to describe the *SiMax* was rephrased in the Method and Material section, as: “The similarity of protein structures between each protein pair was based on the metric called *SiMax* score. *SiMax* score is the normalized heavy atoms RMSD (root mean square deviation) between 2 proteins, which was proposed by CATHEDRAL⁵⁷ and has been used as a standard in the CATH database⁵⁸ to identify ‘structure similar groups’.” These modifications are shown in line 485-488 in page 22.

(2) Can the authors comment on the prevalence of cis- peptide bonds and incorrect amino acid chirality in predictions produced by AlphaFold?

Response: Based on the idea to investigate the prevalence of cis-peptide and incorrect amino acid chirality in AlphaFold predicted structures, we calculate the chirality for each C α atom and the cis/trans-peptide bond (ω -dihedral angle) in all our predicted structures.

For protein chirality, considering all amino acid should adopt a left-handed conformation, we find that AlphaFold has a low error rate on predicted structures (we added a supplemental Fig. S11 to show it, see Fig. R8. attached below). We found that AlphaFold can achieve good accuracy (Right-handed C α rate < 1.007% in A501 and < 2.117% in 3DAC) in most amino acids except cysteine. But AlphaFold have a much higher error rate in cysteine chirality (highest right-handed C α rate 4.25% in A501 and 7.76% in 3DAC across all 5 models and before/after relax). In AlphaFold paper supplemental information page 36, they also mentioned that “AlphaFold can produce almost exactly the chiral pair for the backbone atoms”, which is consistent to our results. Additionally, as Fig. S11 showed that the relax step and setting pLDDT > 70 threshold can have obvious improvement on C α atom chirality (see Fig. R8. attached below). After relax step and pLDDT threshold, the right-handed C α rate for all amino

acid except cysteine is below 0.202‰ in A501 and 0.208‰ in 3DAC, and for cysteine is below 7.016‰ in A501 and 2.037‰ in 3DAC.

For cis-peptide bonds, we compared the cis peptide bond rate to statistics from PDB structure. As shown in added supplemental Fig. S12 (see Fig. R9 attached below), the cis peptide bond rate for amide bond (include Xaa-Xaa and Pro-Xaa) is around 0.1%, while imide bond (include Xaa-Pro and Pro-Pro) is around 4%. According to previous research [Ref: Weiss, Jabs, and Hilgenfeld, “Peptide Bonds Revisited.”], the statistical results of PDB databased showed the cis peptide bond rate for amide bond is 0.028% and imide bond is 5.21%. Our result showed similar cis peptide bond rate propensity to the PDB database.

Above discussions have been added into the revised SI text in the “Discussion about investigate the prevalence of cis-peptide and incorrect amino acid chirality in AlphaFold predicted structures” section.

Fig. R8. (Supplemental Fig. S11.) Chirality analysis of C α in all AlphaFold predicted structures in A501 (A, B) and 3DAC (C, D). Consider all residues C α (except glycine) should adopt a left handed conformation, the wrong chirality count in prediction set and wrong chirality rate is shown in this figure. We also compared the protein structure set before and after pLDDT filter (all models pLDDT > 70)

(A) Wrong chirality count and rate in all predicted A501 structures

(B) Wrong chirality count and rate in pLDDT-filtered A501 structures

(C) Wrong chirality count and rate in all predicted 3DAC structures

(D) Wrong chirality count and rate in pLDDT-filtered 3DAC structures

Fig. R9. (Supplemental Fig. S12.) Cis peptide bond rate analysis (ω -dihedral angle) in all AlphaFold predicted structures in A501 (A) and 3DAC (B). Peptide bonds are classified into 4 categories by its neighbor residues, “Pro” corresponds to proline, and “Xaa” corresponds to all other amino acids.

(A) Cis peptide bond rate of all predicted A501 structures

(B) Cis peptide bond rate of all predicted 3DAC structures

(3) Line 131, can authors check that values are $A501 = 0.31$, $3DAC = 0.67$, and $E. coli = 0.78$ N/m? The sentence beginning Line 132 seems to indicate that the actual values should be assigned as $A501 = 0.78$, $3DAC = 0.67$, and $E. coli = 0.31$, as it says both A501 and 3DAC “showed much higher values than *E. coli*.”

Response: We should apologize for the mistake of values of A501 and *E. coli* at Line 131. The thermal resilience of A501, 3DAC and *E. coli* is 0.78, 0.67 and 0.31 N/m, respectively, which has been corrected in the revised manuscript.

Reviewer #3

The approach followed in this manuscript is in principle quite interesting and may deliver some useful insights. Since my personal expertise lies in the field of scattering techniques I would like to comment only on the neutron scattering part of the manuscript.

Response: we are grateful for the reviewer's encouraging and constructive comments, which will improve the presentation of our work. Line-by-line response is as following:

(i) I think the given values in the manuscript (see page 6) "The mean resilience values of biomolecules in A501, 3DAC and *E. coli* were found to be 0.31, 0.67 and 0.78 N/m, respectively" are not correctly assigned. It would understand if 0.31 goes with *E. coli*??

Response: We apologize that we reversed the values of A501 and *E. coli* by mistake. The thermal resilience of A501, 3DAC, and *E. coli* is 0.78, 0.67, and 0.31 N/m, respectively, which has been corrected in the revised manuscript.

(ii) In the cited paper from Tehei et al. (ref. 22) the authors also measured *E. coli* cells and obtained a mean resilience value of 0.39 N/m. Is this deviation in the error limit of the obtained values? What is the error of the k -values obtained in the presented work? The authors from ref. 22 claimed an error of approx. 5 %. That would mean that the value obtained in this paper is out of the error bound.

Response: We thank the reviewer's comment. *E. coli* used in Tehei's paper (EMBO Rep. 5, 66-70 (2004)) is MRE600, while that used in our work is strain BL21(DE3). Various types of *E. coli* may cause the differences in effective force constants $\langle k \rangle$ of biomacromolecules. Besides, *E. coli* used in our work is grown anaerobically at 37 °C, while *E. coli* used in Tehei's paper is grown in aerobic environment at 37 °C. Different culture conditions may also affect the value of thermal resilience of biomacromolecules in *E. coli*. However, $\langle k \rangle$ of *E. coli* obtained from our work (0.31 N/m) and Tehei's paper (0.39 N/m) are both significantly lower than that of 3DAC (0.67 N/m) and A501 (0.78 N/m), which demonstrates the biomacromolecules in thermophilic/hyperthermophilic prokaryotes is stable at high temperatures. Moreover, we have added the error bar of the resilience estimated from our work. As can be seen in Fig. R10 (added as Fig. 1E in the main text), the error bar is smaller than the difference between the $\langle k \rangle$ values measured in our work and in Tehei's paper. We suspect this difference is real and it should result

from the different bacterial strains and different culture conditions between the two works. The above discussion has been added in the revised SI text.

Fig. R10. (Fig. 1E in main text) Mean macromolecular resilience $\langle k' \rangle$ for *E. coli*, 3DAC and A501. The $\langle k' \rangle$ -values of *E. coli*, 3DAC and A501 is 0.31 ± 0.020 , 0.67 ± 0.015 , 0.78 ± 0.022 N/m, respectively.

Since the method (determination of mean resilience values from different cell types) was to my knowledge not applied in many studies, it is methodically unclear for me how reliable and meaningful the determined values are. Is the difference in k-values between A501 and 3DAC statistically significant? The authors should give more details on the methodically aspects of this approach in order to convince the reader, that the use neutron scattering gives trustable results.

Response: We thank the reviewer's comment. The resilience is determined here as the steepness of the variations of the mean-squared-atomic displacement measured by neutron scattering with temperature. For a simple model of an elastic spring, the so-defined resilience is nothing but the elastic constant, which directly measures the stiffness of the spring. Many research works have used this method to measure the stiffness of protein molecules (Science, 288, 1604-1607 (2000); JACS, 133, 13213-13215 (2011); JPCL, 12, 12402-12410 (2021); Sci. Rep., 6, 37318 (2016);

PCCP., 15, 20951 (2013); Eur. Phys. J. E, 36,78 (2013); Biophys. J., 105, 2157-2165 (2013)) or the average stiffness of the biomolecules inside the cells (Extremophiles, 19, 1099-1108 (2015); J. R. Soc. Interface, 10, 20130003 (2013); EMBO Rep. 5, 66-70 (2004); Eur. Biophys. J., 37, 613-617 (2008); Zeitschrift für Physikalische Chemie, 228, 1121-1133 (2014); Biochem. Biophys. Research Commun., 446, 255-260 (2014); J. Phys. Condens. Matter. 20, 104216 (2008); Sci. Rep. 10:3298 (2020)). We have added the above discussion to the revised main text in line 130-133 in page 6-7.

In the present work, we found that the $\langle k \rangle$ value of A501 is slightly higher than 3DAC, and the difference (0.11 N/m) is statistically meaningful as it is more than five times larger than the error bar (~ 0.02 N/m, see Fig. R10). We note that both A501 and 3DAC have a much larger value than E. coli, which could help the biomacromolecules in A501 and 3DAC adapt to the high-temperature environment they stay with.

(iii)The authors should give information on the q-range of the recorded data.

Response: Thanks for the reviewer's suggestion. The q range for all three cells is from 0.27 to 1.97 \AA^{-1} . We have added this information in the Materials and Methods section in the revised main text.

REVIEWERS' COMMENTS

Reviewer #1 (Remarks to the Author):

The revised manuscript is now an easier read and is now more accessible to those working outside the field. I would suggest briefly considering the comments below before publication.

Additional comments and thoughts after a second read:

-lines 100-101: the references 19-21 should include the new references to protometabolic pathways that were included in the revisions at this point.

-The authors may also wish to include the following papers regarding experimental attempts at protometabolism: [10.1126/sciadv.aav7](https://doi.org/10.1126/sciadv.aav7) and [10.1002/anie.202212932](https://doi.org/10.1002/anie.202212932)

-another question, which the authors might want to briefly address, or not, in the final version of their manuscript, possibly in the conclusion, is this: in the present manuscript, numerous proteins from one bacterial strain and one archaeal strain are compared. Looking forwards, what do the authors expect might be learned beyond what is found in the present paper if they were to compare the metabolic proteomes of numerous bacterial and archaeal strains? Is a big-data approach likely to give different or more robust insights? Is this currently feasible?

Reviewer #2 (Remarks to the Author):

Zhao et al. have modified their manuscript to address each of my concerns with the initial draft. I am happy to recommend publication of the updated manuscript.

Reviewer #3 (Remarks to the Author):

The revised manuscript and the response to the reviewers answered most of my comments satisfactorily. Nevertheless, I have one further comment. The statistical errors of the effective force constants are calculated from a single sample at different temperatures, I assume. Considering these errors, the conclusions (based on the obtained results) made in the manuscript appear reasonable. However, the real error/deviation for a specific species is most probably much larger, since larger deviations appear already for one and the same species, if different strains or different (aerobic versus anaerobic) conditions were used (see the case of *E. coli*). A more convincing approach would be not to employ not only a single sample for a species (as a representative of either a mesophile, or a thermophile or a hyperthermophile) but several samples from different cell culture preparation (and maybe also from different strains) in order to quantify the real spread of values of force constants within a species. Only with the knowledge of this spread one can convincingly confirm the announced correlation between force constants and the thermophilicity. Most probably such an approach is beyond the scope of this manuscript. Unfortunately, the cited original work (which introduced this approach, see ref. 22) did also not analyzed this spread in detail.

Reviewer's Comments:

(point-by-point)

Reviewer #1 (Remarks to the Author):

The revised manuscript is now an easier read and is now more accessible to those working outside the field. I would suggest briefly considering the comments below before publication.

Additional comments and thoughts after a second read:

-lines 100-101: the references 19-21 should include the new references to protometabolic pathways that were included in the revisions at this point.

Response: We thank the reviewer for the carefully reading. All references including the additional papers below provided by the reviewer have been cited in line 101 page 5.

-The authors may also wish to include the following papers regarding experimental attempts at protometabolism: 10.1126/sciadv.aav7 and 10.1002/anie.202212932

Response: We thank the additional reference provided by the reviewer. These two publications have been included in the citation for the description of the consistency between the structure conserved metabolic modules and the prebiotic chemistry confirmed protometabolism, in line 101 page 5, line 356 page 17, and legend of Fig. 5.

-another question, which the authors might want to briefly address, or not, in the final version of their manuscript, possibly in the conclusion, is this: in the present manuscript, numerous proteins from one bacterial strain and one archaeal strain are compared. Looking forwards, what do the authors expect might be learned beyond what is found in the present paper if they were to compare the metabolic proteomes of numerous bacterial and archaeal strains? Is a big-data approach likely to give different or more robust insights? Is this currently feasible?

Response: We appreciate the constructive suggestion from the reviewer. To address the question raised by the reviewer and make a perspective on big-data based structure predictions on numerous bacterial and archaeal strains, we added the following sentences at the end of the end of the Discussion: "Although only a relatively small number of structures have been solved in

this study compared to the incredibly diverse prokaryotic proteins, our results proposed that high-throughput protein 3D structure prediction is a promising approach in ancestral investigations. In the future, if we can combine the vast amount of microbial (meta)genomic information from diverse environments (especially extreme environments similar to early Earth conditions) accumulated over the past decade, high-throughput protein structure prediction could help us summarize the conserved structural and metabolic modules inherited from ancestors in modern cells in a broader context. These structure-based ancestral modules could build a bridge to decipher the key factors that replaced prebiotic native metabolism with enzyme-catalyzed life processes in the origin of life.” in line 367-387 page 18.

Reviewer #2 (Remarks to the Author):

Zhao et al. have modified their manuscript to address each of my concerns with the initial draft. I am happy to recommend publication of the updated manuscript.

Response: We are very grateful to the reviewer for the help and recognition.

Reviewer #3 (Remarks to the Author):

The revised manuscript and the response to the reviewers answered most of my comments satisfactorily. Nevertheless, I have one further comment. The statistical errors of the effective force constants are calculated from a single sample at different temperatures, I assume. Considering these errors, the conclusions (based on the obtained results) made in the manuscript appear reasonable. However, the real error/deviation for a specific species is most probably much larger, since larger deviations appear already for one and the same species, if different strains or different (aerobic versus anaerobic) conditions were used (see the case of *E. coli*). A more convincing approach would be not to employ not only a single sample for a species (as a representative of either a mesophile, or a thermophile or a hyperthermophile) but several samples from different cell culture preparation (and maybe also from different strains) in order to quantify the real spread of values of force constants within a species. Only with the knowledge of this spread one can convincingly confirm the announced correlation between force constants and the thermophilicity. Most probably such an

approach is beyond the scope of this manuscript. Unfortunately, the cited original work (which introduced this approach, see ref. 22) did also not analyzed this spread in detail.

Response: We thank the reviewer's comment. In this work, we compared the thermal resilience ($\langle k' \rangle$) of biomacromolecules obtained from the A501, 3DAC, and *E. coli* grown at their respective physiological temperatures (85, 75, and 37 °C) to illustrate that biomacromolecules in A501 and 3DAC are structurally more stable than those in *E. coli*. We note that the comparison of these cells at their respective physiological conditions is of direct biological relevance, and that is why we did so. However, we agreed with the reviewer that the value of $\langle k' \rangle$ of a cell could differ when the preparation condition changes. Here, we collected the $\langle k' \rangle$ values of biomacromolecules in *E. coli* grown in different conditions, including the anaerobic environment, in D₂O, in an unstressed condition, and in heat-shocked condition, which is 0.31, 0.19, 0.42, and 0.30 N/m (Eur. Biophys. J., 37, 613-617 (2008).; EMBO Rep. 5, 66-70 (2004)), respectively. As a result, the resulting value when averaged over different preparation conditions is 0.31 ± 0.09 N/m. Although the spread among different conditions is larger than that obtained from a single condition, one still can deduce that the resilience of biomolecules in *E. coli* is significantly smaller than those in 3DAC and A501 (0.67 and 0.78 N/m), i.e., the latter two are structurally more stable. Moreover, 3DAC and A501 grow extremely slowly at conditions other than the physiological temperature or even die in the aerobic culture, thereby the weight of cells cannot reach the requirement for dynamic neutron scattering measurements. Thus, we cannot do a similar statistical analysis on these two cells. The above discussion has been added to the revised SI Note.